# Breaking the Block: Preserving Data Continuity to Train Superior SAEs for Instruct Models

Jiaming Li [* 1 2]  Haoran Ye [* 3]  Yukun Chen [1 2]  Xinyue Li  Lei Zhang [1 2]  Hamid Alinejad-Rokny [4]
Jimmy Chih-Hsien Peng [† 3]  Min Yang [† 1 5]

## Abstract

Sparse Autoencoders (SAEs) are a cornerstone of mechanistic interpretability. Existing training methods inherit the Block Training paradigm from LLM pre-training, which introduces destructive gradient noise in instruct models due to attention leakage from unrelated contexts. Using GSNR analysis, we theoretically characterize this issue and propose **F**inetuning-**a**ligned **S**equential **T**raining (*FAST*), a sequential training paradigm specifically designed for instruct models. *FAST* aligns SAE training with the data distribution and activation patterns of instruct models, substantially improving both reconstruction fidelity and feature interpretability. Experimental results show that *FAST* achieves higher GSNR, a significantly lower log-scaled MSE of 0.6468 compared to the baseline's 5.1985, and a near-zero Delta Loss (-0.51% to 0.37%). Moreover, on Llama-3.2-3B-it, *FAST* produces 21.1% high-quality features, substantially outperforming baseline methods that achieve 7.0% and 10.2%. We further find that intervening on special token activations through SAEs can improve generation quality, revealing new opportunities for fine-grained control. Our codes are available as open source at https://github.com/Geaming2002/FAST.

## 1. Introduction

Large Language Models (LLMs) demonstrate exceptional performance across diverse natural language processing

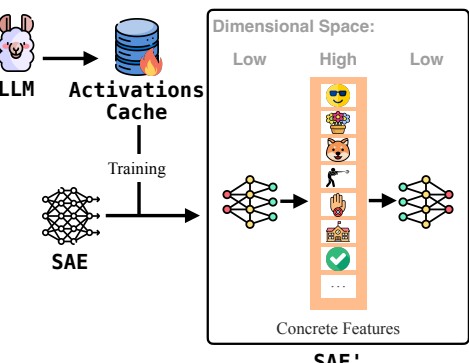

*Figure 1.* Overview of the sparse autoencoder for interpreting internal representations of large language models.

tasks (Brown et al., 2020; Ouyang et al., 2022; Guo et al., 2025). As these models increasingly permeate critical applications, understanding their internal mechanisms has become imperative for ensuring alignment and safety (Bengio et al., 2024; Bereska & Gavves, 2024; Ji et al., 2025). Sparse autoencoders (SAEs) serve as a powerful tool for interpreting LLMs by mapping high-dimensional activations to sparse, interpretable feature spaces (Bereska & Gavves, 2024; Bricken et al., 2023; Huben et al., 2024). SAE training shown in Figure 1 employs dictionary learning to enforce sparsity, consistent with the linear representation and superposition hypotheses (Elhage et al., 2022; Arora et al., 2018). While initial research predominantly focused on pre-trained base models, the community's attention is rightfully shifting towards instruct models, which directly interact with users and execute complex directives.

However, we identify a fundamental oversight in the current landscape of SAE training for instructing models. Large-scale initiatives, such as Gemma Scope (Lieberum et al., 2024) and Llama Scope (He et al., 2024), tend to inherit the Block Training (*BT*) paradigm from the LLM pre-training phase. In this paradigm, documents are concatenated and arbitrarily split into fixed-length sequences (e.g., 2048 or 4096 tokens) (Joseph Bloom & Chanin, 2024; Bricken et al., 2023) to train SAE as shown in Figure 3(b). While effective for base models, this method faces significant limitations

---

[1]Shenzhen Institutes of Advanced Technology, Chinese Academy of Sciences, Shenzhen, China [2]University of Chinese Academy of Sciences, Beijing, China [3]National University of Singapore, Singapore [4]The University of New South Wales, Sydney, Australia [5]Shenzhen University of Advanced Technology, Shenzhen, China. Correspondence to: Jimmy Chih-Hsien Peng <jpeng@nus.edu.sg>, Min Yang <min.yang@siat.ac.cn>.

*Proceedings of the 43$^{rd}$ International Conference on Machine Learning*, Seoul, South Korea. PMLR 306, 2026. Copyright 2026 by the author(s).

when applied to instruct models (Joseph Bloom & Chanin, 2024). The semantic discontinuity from combining diverse sources undermines semantic coherence for downstream tasks, degrading SAE performance.

We diagnose this mismatch as a source of destructive interference that fundamentally degrades interpretability. Utilizing GSNR analysis (Liu et al., 2020), we demonstrate that *BT* induces Attention Contamination—leakage from unrelated predecessor documents. In instruct models, this acts as high-variance gradient noise that obscures precise instruction-response mappings.

To correct the paradigm, we propose **F**inetuning-**a**ligned **S**equential **T**raining (*FAST*). *FAST* aligns the SAE training stream with the Supervised Fine-Tuning (SFT) objective. This approach eliminates attention contamination, ensuring that gradient updates are driven solely by valid causal signals. By preserving the semantic integrity of instruction-response pairs, *FAST* stabilizes the semantic space and significantly enhances the quality of feature extraction. Experimental results validate the efficacy of *FAST*. GSNR analysis confirms improved training stability, with *FAST* doubling the signal-to-noise ratio (0.50 vs. 0.26) by minimizing gradient noise. This stability yields superior reconstruction fidelity: *FAST* achieves a log-scaled MSE of 0.6468 (significantly outperforming the baseline's 5.1985) and maintains Delta Loss near zero (-0.51% to 0.37%). Consequently, feature quality is markedly enhanced; on Llama-3.2-3B-it, *FAST* yields 21.1% high-quality features, surpassing the 7.0% and 10.2% achieved by baselines. Additionally, SAEs are used to study special token impacts on outputs, offering insights into their roles and applications.

Our contributions are:

- Utilizing Gradient Signal-to-Noise Ratio analysis, this study reveals that *BT* introduces a critical distributional mismatch and generates destructive gradient noise via attention contamination.

- We propose Finetuning-aligned Sequential Training (*FAST*), a novel method for training SAEs on instruct models which significantly improves SAE performance on token reconstruction and feature interpretability.

- We utilize SAEs to investigate special token influence on model outputs, providing new insights for practical applications.

## 2. Related Work

**Sparse Autoencoders for LLM.** Mechanistic interpretability aims to address the "black box" nature of LLMs by reverse-engineering their internal representations (Nanda, 2022; 2023). Sparse Autoencoders (SAEs) have emerged

as a key tool in this domain, decomposing polysemantic neurons into interpretable, sparse features through dictionary learning (Bricken et al., 2023; Bereska & Gavves, 2024). While standard SAEs have shown promise, several recent methods aim to improve interpretability and adaptability. Routing-enhanced SAEs (e.g., RouteSAE) introduce expert-style routing to adapt feature usage and reduce interference (Shi et al., 2025). Sparsity-and-isolation-focused frameworks (e.g., SAIF) strengthen feature isolation to promote monosemanticity (He et al., 2025). Low-rank adaptations have also been explored to lower compute costs while improving transfer (Chen et al., 2025).

Due to space limitations, we provide a detailed discussion on the theoretical foundations of SAEs (including the linear representation and superposition hypotheses) and the broader context of mechanistic interpretability in Appendix A. Existing approaches like RouteSAE (Shi et al., 2025) and SAIF (He et al., 2025) are largely orthogonal to training-data and optimization alignment. Our Finetuning-aligned Sequential Training (*FAST*) targets this dimension by preserving instance-level boundaries and aligning SAE training with fine-tuning objectives.

## 3. Theoretical Analysis: Noise and Alignment

In this section, we provide a theoretical analysis of the detrimental effects of Block Training (*BT*) on instruct model SAEs using the framework of Gradient Signal-to-Noise Ratio (GSNR) (Liu et al., 2020). We model SAE training as dictionary learning over a distribution $\mathcal{D}$.

### 3.1. Gradient Noise of Semantic Discontinuity

We first formalize the gradient structure under Block Training, focusing on the "Attention Contamination" phenomenon in Figure 2.

**Definition 3.1** (Gradient Under Attention Contamination). Let $S_{BT} = [d_{pre}; d_{cur}]$ be a concatenated sequence, where $d_{pre}$ and $d_{cur}$ are semantically unrelated. For tokens in the current instance $d_{cur}$, causal attention under Block Training allows attention to preceding tokens in $d_{pre}$. Since Softmax attention assigns strictly positive weights to unmasked positions with finite logits, the representations of $d_{cur}$ may be affected by the unrelated predecessor context.

Let $g_{BT}(\theta)$ denote the stochastic gradient computed from $d_{cur}$ under the concatenated context $[d_{pre}; d_{cur}]$, and let $g_{signal}(\theta)$ denote the clean stochastic gradient obtained when $d_{cur}$ is processed independently. We define the attention-contamination term as

$$\epsilon_{poll}(\theta) := g_{BT}(\theta) - g_{signal}(\theta). \quad (1)$$

Equivalently,

$$g_{BT}(\theta) = g_{signal}(\theta) + \epsilon_{poll}(\theta). \quad (2)$$

### *Block Training*: Attention Contamination

Input: [$d_{pre}$ (Unrelated History)] [$d_{cur}$ (Current Document)]

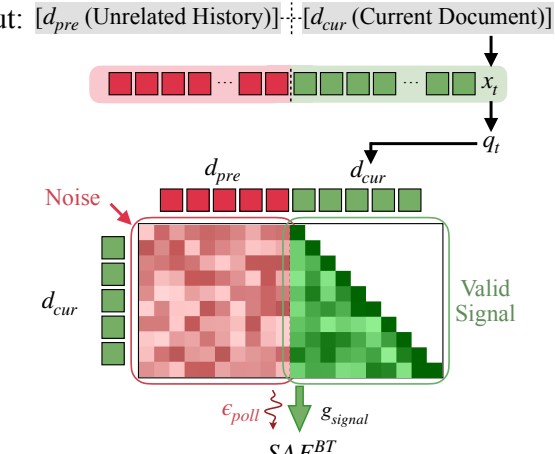

### *FAST*: Clean Causal Attention

Input: [$d_{cur}$ (Current Document)]

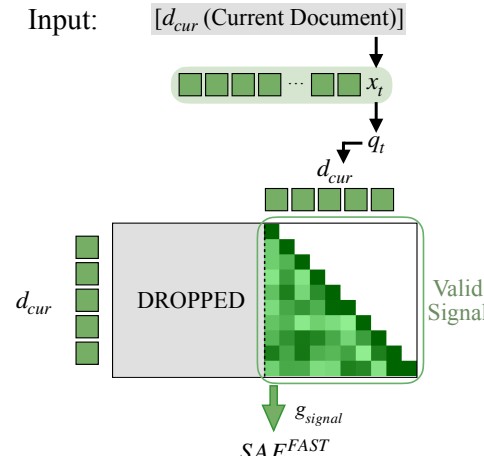

*Figure 2.* Visualization mechanism of gradient noise induced by attention contamination. **Left**: In Block Training, the concatenation of unrelated documents ($d_{pre}$ and $d_{cur}$) introducing $\epsilon_{poll}$ into the gradient. **Right**: *FAST* processes documents independently, effectively "drop" the unrelated history. This enforces a clean causal attention mask, ensuring the gradient consists solely of the valid signal $g_{signal}$.

Here, $\epsilon_{poll}(\theta)$ captures the gradient perturbation induced by cross-instance attention leakage from the unrelated predecessor $d_{pre}$.

To analyze the variance effect of this perturbation, we introduce a standard orthogonal-noise modeling assumption. All expectations below are taken over the sampling randomness of stochastic gradient estimation.

**Assumption 3.2** (Orthogonality of Semantic Noise). Following the standard stochastic-gradient noise modeling framework in which the observed gradient is decomposed into a clean component and a perturbation term (Bottou et al., 2018; Mandt et al., 2017), we model the interference term $\epsilon_{poll}$ as an approximately zero-mean perturbation that is uncorrelated with the clean gradient:

$$\mathbb{E}[\epsilon_{poll}(\theta)] \approx 0, \tag{3}$$

and

$$\mathbb{E}\left[\langle g_{signal}(\theta) - \mathbb{E}[g_{signal}(\theta)], \epsilon_{poll}(\theta) - \mathbb{E}[\epsilon_{poll}(\theta)]\rangle\right] \approx 0. \tag{4}$$

This assumption is used to isolate the variance effect of attention contamination. If it is violated, the contamination may additionally introduce bias or covariance effects.

Based on this assumption, we derive the variance structure of the gradient. Based on this assumption, we derive the variance structure of the gradient.

**Lemma 3.3** (Variance Decomposition). *For a vector-valued random variable $X$, define the total vector variance as*

$$\mathrm{Var}(X) := \mathbb{E}\left[\|X - \mathbb{E}[X]\|_2^2\right]. \tag{5}$$

*Under Assumption 3.2, the variance of the Block Training gradient decomposes as*

$$\mathrm{Var}(g_{BT}) \approx \sigma_{signal}^2 + \sigma_{poll}^2, \tag{6}$$

*where $\sigma_{signal}^2 := \mathrm{Var}(g_{signal})$ and $\sigma_{poll}^2 := \mathrm{Var}(\epsilon_{poll})$.*

*Proof.* Using $g_{BT} = g_{signal} + \epsilon_{poll}$ and the definition of total vector variance, we have

$$\begin{aligned}
\mathrm{Var}(g_{BT}) &= \mathrm{Var}(g_{signal} + \epsilon_{poll}) \\
&= \mathrm{Var}(g_{signal}) + \mathrm{Var}(\epsilon_{poll}) \\
&\quad + 2\mathbb{E}\left[\langle g_{signal} - \mathbb{E}[g_{signal}], \epsilon_{poll} - \mathbb{E}[\epsilon_{poll}]\rangle\right].
\end{aligned} \tag{7}$$

By Assumption 3.2, the cross term is approximately zero. Therefore,

$$\mathrm{Var}(g_{BT}) \approx \sigma_{signal}^2 + \sigma_{poll}^2. \tag{8}$$

$\square$

We can now state our main result comparing Block Training to a clean-context ideal, where the contamination term is absent.

**Theorem 3.4** (GSNR Degradation in Block Training). *Let $g(\theta)$ be a stochastic gradient estimator. We define its Gradient Signal-to-Noise Ratio (GSNR) as*

$$\mathrm{GSNR}(g) := \frac{\|\mathbb{E}[g(\theta)]\|_2^2}{\mathrm{Var}(g(\theta))}, \tag{9}$$

*where $\mathrm{Var}(\cdot)$ denotes the total vector variance. Under Assumption 3.2, let $\mu_{signal} := \mathbb{E}[g_{signal}(\theta)]$. If $\|\mu_{signal}\|_2 >$*

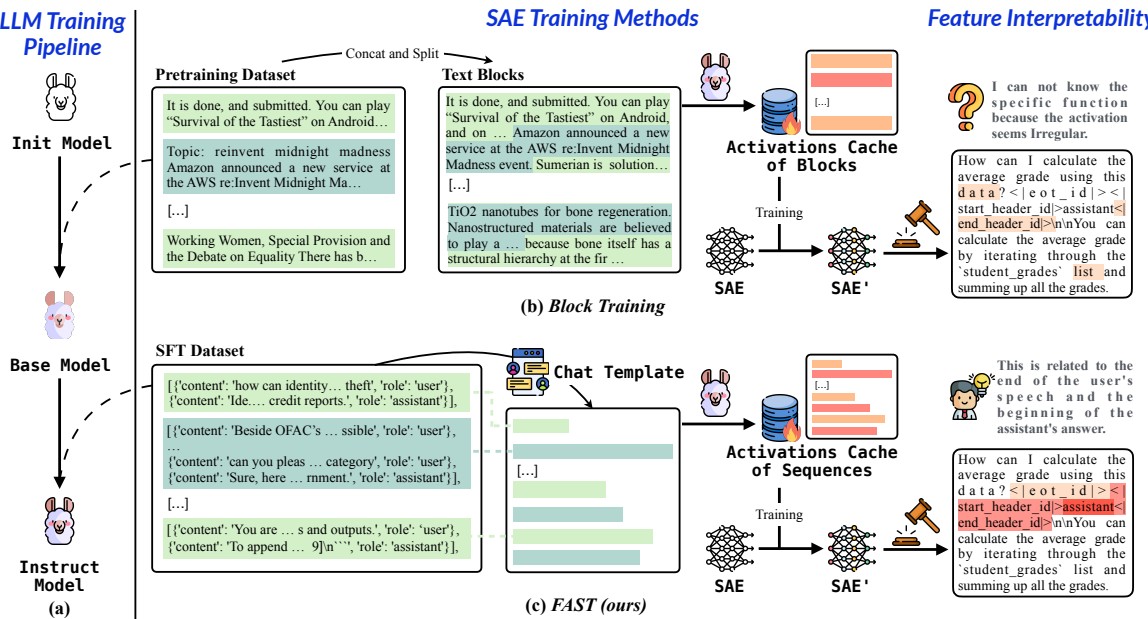

*Figure 3.* Illustration of the LLM training pipeline and SAE training methods. **(a)** The pipeline transitions from pretraining to fine-tuning. **(b)** Block Training (*BT*) concatenates datasets. **(c)** Finetuning-aligned Sequential Training (*FAST*) processes data instances independently.

$0$, $\sigma^2_{signal} > 0$, and $\sigma^2_{poll} > 0$, then Block Training degrades the GSNR relative to the clean-context ideal:

$$\text{GSNR}_{BT} \approx \frac{\|\mu_{signal}\|^2_2}{\sigma^2_{signal} + \sigma^2_{poll}} < \frac{\|\mu_{signal}\|^2_2}{\sigma^2_{signal}} = \text{GSNR}_{ideal} .$$

(10)

*Proof.* Under Assumption 3.2, the interference term is approximately zero-mean and uncorrelated with the clean gradient. Therefore, $\mathbb{E}[g_{BT}(\theta)] \approx \mathbb{E}[g_{signal}(\theta)] = \mu_{signal}$, while Lemma 3.3 gives $\text{Var}(g_{BT}) \approx \sigma^2_{signal} + \sigma^2_{poll}$. In the clean-context ideal, the contamination term is absent, so the variance reduces to $\sigma^2_{signal}$. Since $\sigma^2_{poll} > 0$, $\sigma^2_{signal} > 0$, and $\|\mu_{signal}\|_2 > 0$, we obtain

$$\frac{\|\mu_{signal}\|^2_2}{\sigma^2_{signal} + \sigma^2_{poll}} < \frac{\|\mu_{signal}\|^2_2}{\sigma^2_{signal}}.$$

(11)

This completes the proof. $\square$

This GSNR degradation suggests a penalty on optimization efficiency. Appendix B further quantifies this effect through a standard non-convex SGD upper-bound analysis and derives a variance-dependent efficiency ratio, further motivating our proposed method.

The divergent efficacy of *BT* in base versus instruct models stems from distinct topological properties of their data manifolds. Base model corpora, such as web text, are characterized by high entropy and weak structural constraints. In this context, attention leakage may function as a form of

implicit regularization, aiding the learning of broad correlations. Conversely, instruct models operate in a low-entropy, precision-critical regime. Given the strict causal dependencies required for instruction following, noise from unrelated history acts not as regularization but as destructive interference. This obscures instruction-response mappings, causing the SAE to encode spurious, noise-dependent features and severely degrading interpretability.

## 3.2. Motivation of *FAST*

Current approaches predominantly adopt Block Training (*BT*) for SAEs. However, our analysis in Appendix B exposes a critical deficiency in this paradigm. As a result, the attention contamination in *BT* imposes a lower efficiency to train SAEs as shown in Equation 39. In instruction tuning, where signal variance $\sigma^2_{signal}$ is naturally low (sparse features), this penalty $\rho$ becomes prohibitively large, necessitating a more efficient training approach.

To address this, we propose Finetuning-aligned Sequential Training (*FAST*), as depicted in Figure 3(c).By processing instances independently, *FAST* eliminates the influence of unrelated contexts, effectively reducing the interference variance to zero ($\sigma^2_{poll} = 0$).This restoration of semantic integrity maximizes the GSNR, as validated by our theoretical analysis.Unlike *BT*, which degrades feature quality through gradient noise, *FAST* aligns with the specific optimization landscape of fine-tuning, enabling the SAE to capture nuanced, task-specific representations effectively.

# 4. Finetuning-aligned Sequential Training

This section elaborates on *FAST* detailing how the framework effectively aligns sequential training dynamics with fine-tuning objectives to ensure efficiency.

## 4.1. Data Processing

As previously described, *FAST* trains the SAE using fine-tuning datasets. Specifically, multiple multi-turn dialogue datasets are collected, and each data instance is combined with the corresponding chat template of the instruct model. This process not only introduces special tokens but also ensures consistency with the data processing methodology used during the fine-tuning phase of the model.

A key innovation lies in independent processing of each data instance, rather than concatenating multiple instances before inputting them into the model. By eliminating the fixed-length block requirement of Block Training, the dataset is processed sequentially. Each data instance is processed as an independent sequence to extract hidden layer activations for SAE training, as illustrated in Figure 3(c). This approach effectively avoids semantic discontinuity caused by data concatenation, while preserving the semantic integrity of each instance, thereby providing higher-quality inputs for training the SAE.

## 4.2. SAE

We analyze the residual stream using two SAE variants: the **Standard ReLU-based SAE** (Bricken et al., 2023) and the **JumpReLU SAE** (Rajamanoharan et al., 2024b). Both architectures project an input $\mathbf{x} \in \mathbb{R}^d$ to a sparse latent $\mathbf{z}$ and reconstruct it as $\hat{\mathbf{x}}$. The forward pass is:

$$\mathbf{u} = \mathbf{W}^{\text{enc}}\mathbf{x} + \mathbf{b}^{\text{enc}}, \quad \hat{\mathbf{x}} = \mathbf{W}^{\text{dec}}f(\mathbf{u}) + \mathbf{b}^{\text{dec}}. \quad (12)$$

Initialization protocols and detailed definitions are in Appendix C and Appendix E.

**Standard ReLU SAE.** This model uses the ReLU activation. To induce sparsity, it minimizes a loss $\mathcal{L}_{\text{std}}$ combining reconstruction error with an $L_1$ penalty. However, the $L_1$ term introduces shrinkage bias, reducing active feature magnitudes:

$$f(\mathbf{u}) = \text{ReLU}(\mathbf{u}), \quad (13)$$

$$\mathcal{L}_{\text{std}} = \|\mathbf{x} - \hat{\mathbf{x}}\|_2^2 + \lambda\|\mathbf{z}\|_1. \quad (14)$$

**JumpReLU SAE.** The JumpReLU SAE employs a threshold-based activation $f(\mathbf{u}) = \mathbf{u} \odot H(\mathbf{u} - \theta)$, where $\theta$ is a learnable threshold and $H(\cdot)$ is the Heaviside step function. The training objective effectively approximates the $L_0$ norm:

$$\mathcal{L}_{\text{jump}} = \|\mathbf{x} - \hat{\mathbf{x}}\|_2^2 + \lambda\|\mathbf{z}\|_0. \quad (15)$$

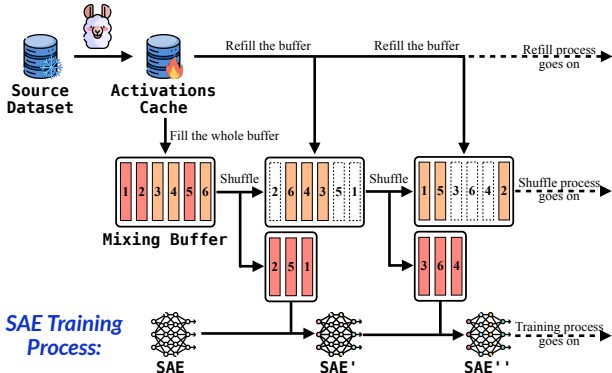

*Figure 4.* The mixing activation buffer is shuffled, half is sent to the SAE for training, and the resulting new activations are used to refill the buffer. This iterative process ensures data diversity and storage efficiency.

This formulation suppresses noise below $\theta$ without penalizing the scale of significant features.

## 4.3. Mixing Activation Buffer

Activation values, which represent the activation levels of hidden layer dimensions during the model's forward pass, require significant storage space. To mitigate this challenge, we employ a producer-consumer framework inspired by previous studies (Joseph Bloom & Chanin, 2024), wherein the LLM generates activations and stores them in a dedicated buffer.

As shown in Figure 4, the process begins with the buffer being filled to capacity with activation values. Once the buffer is full, the activations are shuffled to ensure randomness and diversity. Subsequently, half of the shuffled activations are sent to the SAE model for training, while the other half remains in the buffer. After training, the buffer is replenished with new activations generated by the model, and the cycle repeats. This iterative mechanism optimizes storage efficiency and ensures a high level of data variability, thereby enhancing the robustness of model training. By leveraging the mixing buffer, this approach effectively balances data diversity with storage efficiency.

# 5. Experiments

## 5.1. Experiment Setup

**Dataset.** We construct a large-scale instruction dataset for fine-tuning LLMs by combining several publicly available, high-quality datasets, including WildChat-1M-Full (Zhao et al., 2024), Infinity-Instruct (Li et al., 2025), tulu-3-sft-mixture (Lambert et al., 2025), orca-agentinstruct-1M-v1-cleaned [1], and lmsys-chat-1m (Zheng et al., 2023). After

---

[1] https://huggingface.co/datasets/mlabonne/orca-agentinstruct-1M-v1-cleaned

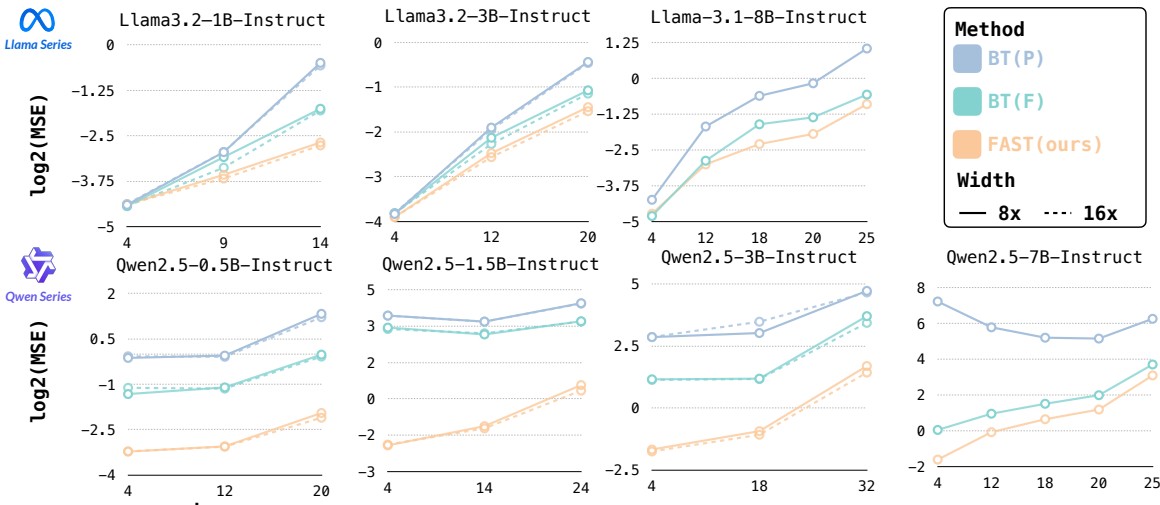

*Figure 5.* $\text{MSE}_{st}$ performance of the JumpReLU SAE (all metrics are presented in log scale, where lower values indicate better SAE reconstruction performance). Within the JumpReLU architecture, *FAST* exhibits the best reconstruction capability compared to *BT(P)* and *BT(F)*.

applying a 20-gram deduplication strategy, it is reduced to 4,758,226 samples. Details are in Appendix D.

**LLMs.** We conduct experiments on seven models from two families: Llama (Llama-3.1, Llama-3.2) (Grattafiori et al., 2024) and Qwen (Qwen-2.5) (Qwen et al., 2025), selected for their state-of-the-art performance to evaluate our approach's robustness and generalization across families and scales. The models and their respective layer configurations, detailed in Table 5, are selected from various depths to mitigate depth bias. Following prior works (Bereska & Gavves, 2024; Bricken et al., 2023; Gao et al., 2024), we train SAEs on the residual stream, as inter-layer relationships have minimal impact on performance.

**Baselines.** Prior to this study, all SAE model training methods exclusively utilize the Block Training (*BT*) strategy. Depending on the type of training dataset used, Block Training can be categorized into two primary forms: *BT(P)* and *BT(F)* as follows:

- *BT(P)*: Block Training using the pretraining dataset. The pretraining dataset is processed by concatenating and segmenting the data into text blocks of equal length, which are then used for training the SAE model.

- *BT(F)*: Block Training using the finetuning dataset. This approach utilizes a finetuning dataset. The data within the dataset is concatenated to form text blocks.

For *BT(P)*, we utilize the pile-uncopyrighted dataset [2]. As

for *BT(F)*, we use the finetuning dataset mentioned before which is also used in *FAST*.

[2] https://huggingface.co/datasets/monology/pile-uncopyrighted

**Configuration.** SAEs are trained on 8*NVIDIA A100 GPUs using `sae_lens` (Joseph Bloom & Chanin, 2024) with custom implementation. For models with more than 7B parameters, the expansion factor of SAE is fixed at 8X, whereas for other models, the expansion factor can be 8X or 16X. To ensure fairness across methods at the same data scale, the number of training tokens is set to 40.96M. For *BT(P)* and *BT(F)*, `context_size` is 2,048, with each text block containing 2,048 tokens. For *FAST*, no explicit `context_size` is required; instead, a truncation length of 8,192 is applied to manage memory usage. For JumpReLU SAE, $L_{\text{sparsity}}$ is 0.01, while for Standard SAE, it is 5. Further parameter details are in Appendix F.

To assess whether the mismatch between *FAST* (truncation 8192) and *BT* (`context_size` 2048) could confound results, we computed the length distribution of *FAST* training instances. As shown in Table 4, 96.6% of instances are $\leq$ 2048 tokens, suggesting that effective contexts are largely comparable; thus differences are primarily due to the training paradigm.

**Evaluation Metric.** While GSNR serves as our primary theoretical diagnostic to characterize the gradient noise induced by Block Training, we further employ three complementary metrics to comprehensively evaluate SAE performance across different aspects: reconstruction quality, behavioral preservation, and feature interpretability:

**Mean Squared Error (MSE).** The performance of the SAE is assessed using the Mean Squared Error (MSE), which is

*Table 1.* GSNR comparison during early training (Steps 0–100 with interval 10). This table supplements the summarized results in the main text, providing granular details on the reconstruction stability of different methods.

| Models | Method | Training Step | | | | | | | | | | | Win Rate |
|---|---|---|---|---|---|---|---|---|---|---|---|---|---|
| | | 0 | 10 | 20 | 30 | 40 | 50 | 60 | 70 | 80 | 90 | 100 | |
| Qwen2.5-7B-Instruct | *BT(P)* | 0.22 | 0.17 | 0.22 | 0.20 | 0.17 | 0.22 | 0.21 | 0.20 | 0.20 | 0.20 | 0.19 | |
| | *BT(F)* | 0.23 | 0.19 | 0.17 | 0.22 | 0.22 | 0.18 | 0.17 | 0.19 | 0.20 | 0.19 | 0.19 | 0.99 |
| | *FAST* | **0.43** | **0.50** | **0.52** | **0.53** | **0.50** | **0.31** | **0.30** | **0.33** | **0.33** | **0.31** | **0.28** | |
| Llama-3.1-8B-Instruct | *BT(P)* | 0.26 | 0.26 | 0.26 | 0.26 | 0.25 | 0.24 | 0.21 | 0.22 | 0.24 | 0.24 | 0.24 | |
| | *BT(F)* | 0.24 | 0.24 | 0.24 | 0.24 | 0.24 | 0.24 | 0.23 | 0.23 | 0.23 | 0.23 | 0.23 | 1.00 |
| | *FAST* | **0.50** | **0.45** | **0.41** | **0.45** | **0.46** | **0.40** | **0.39** | **0.37** | **0.36** | **0.36** | **0.37** | |

calculated as:

$$\text{MSE} = \frac{\sum_{i=1}^{N} \frac{1}{L_i} \sum_{j=1}^{L_i} \sum_{k=1}^{H} (y_{i,j,k} - \hat{y}_{i,j,k})^2}{N \cdot H} \quad (16)$$

where $N$ denotes the size of the dataset, $L_i$ represents the length of the $i$-th sequence, and $H$ refers to the hidden dimension of the model. To evaluate the SAE's performance specifically on special tokens, we also compute the MSE of special tokens, denoted as $\text{MSE}_{st}$[3]. Lower MSE values reflect better model performance.

**Delta Loss.** This metric measures the cross-entropy loss difference before and after applying the SAE to the residual stream, indicating how well the original model behavior is preserved. Delta loss approaching 0 indicates better preservation, reflecting the SAE's ability to accurately reconstruct activations without distortions. Formally:

$$\Delta L = L_{\text{CE}}^{\text{SAE}} - L_{\text{CE}}^{\text{original}} \quad (17)$$

where $L_{\text{CE}}^{\text{SAE}}$ represents the model's cross-entropy loss with SAE reconstructions at the target layer, and $L_{\text{CE}}^{\text{original}}$ is the original model's loss without intervention. Lower absolute values indicate better behavioral preservation.

**KL Divergence.** To assess behavioral fidelity, we measure the KL Divergence between the original distribution $P$ and the SAE-reconstructed distribution $Q$. A value approaching 0 implies that the SAE preserves the model's output distribution effectively. Formally:

$$D_{\text{KL}}(P \parallel Q) = \sum_{x \in \mathcal{V}} P(x) \log \left( \frac{P(x)}{Q(x)} \right) \quad (18)$$

where $\mathcal{V}$ denotes the vocabulary space, $P(x)$ is the probability distribution over tokens generated by the original model, and $Q(x)$ is the distribution produced by the model with SAE-reconstructed activations. Unlike Delta Loss, KL Divergence provides a holistic assessment of semantic preservation, penalizing any deviations in the model's full probabilistic output.

---

[3]To facilitate a more direct comparison of performance across different methods, all MSE values are transformed using $\log_2$.

## 5.2. Main Results

A random sample of 5,000 dialogues is extracted from the remaining dataset for evaluation in MSE and Delta Loss metrics. Figure 5 compares the $\text{MSE}_{st}$ scores of three methods using the JumpReLU SAE, while Figure 7 illustrates the $\text{MSE}_{st}$ performance of the Standard SAE. Detailed results for both MSE and $\text{MSE}_{st}$ are presented in Appendix G. Additionally, to demonstrate the generalizability of *FAST* on domain-specific models, we further evaluate our method on Qwen2.5-Math-7B, with detailed results provided in Appendix G.8.

In terms of overall token reconstruction (MSE), the JumpReLU architecture with Qwen models demonstrates similar patterns, with *FAST* consistently outperforming baseline methods. *FAST* achieves superior performance across most configurations. For instance, in Llama-3.2-3B-Instruct-L20-8X-Standard, *FAST* attains -0.9527, significantly surpassing the baselines which score -0.6926 and -0.9186. In special token reconstruction ($\text{MSE}_{st}$), *FAST* shows marked improvements across models. In Qwen2.5-7B-Instruct-L18-8X-Standard, *FAST* achieves 0.6468, outperforming the baselines (5.1985 and 1.5093). In the JumpReLU SAEs, it achieves -9.7604 compared to -4.0005 and -8.0743. For Delta Loss (Table 14), *FAST* consistently achieves values closer to zero (-0.51% to 0.37%, most within ±0.1%) compared to *BT(P)* and *BT(F)* methods showing larger deviations up to 51.43% and 41.19% respectively. To assess distributional shifts, we compute the KL Divergence across the vocabulary (see Appendix I). The results confirm that *FAST* achieves significantly lower divergence than baselines, indicating effective mitigation of information loss and better preservation of the model's behavior.

Overall, *FAST* consistently outperforms baseline methods across all evaluation metrics. It excels in both general and special token reconstruction, achieves Delta Loss values closest to zero (ranging from -0.51% to 0.37%) and minimal KL divergence, demonstrating superior performance with substantial improvements across all tested configurations. Furthermore, *FAST* significantly reduces training duration

*Table 2.* Ablation study on context length using Llama-3.1-8B-Instruct (Layer 20, 8X, JumpReLU SAE). *FAST* maintains its significant performance advantages over the baseline even when strictly constrained to the same 2048-token limit.

| Method | $\log_2(\text{MSE})$ | $\log_2(\text{MSE}_{st})$ | KL | $\text{KL}_{st}$ |
|---|---|---|---|---|
| *BT(F)* | -14.80 | -10.45 | 8.19e-05 | 1.02e-03 |
| *FAST-2048* | -15.42 | -13.02 | 2.62e-05 | 5.73e-04 |
| *FAST-8192* | -15.56 | -13.15 | 2.59e-05 | 5.69e-04 |

compared to baselines (see Appendix K). These results establish *FAST* as a robust and efficient training method for Sparse Autoencoders.

### 5.3. GSNR Results

To validate the training stability, we assess the Gradient Signal-to-Noise Ratio (GSNR) during the critical early phase. As shown in Table 1, *FAST* consistently yields significantly higher GSNR values compared to both Block Training baselines across all evaluated models. For instance, in the Llama-3.1-8B model, *FAST* doubles the baseline GSNR (0.50 vs 0.26).This result suggests that by preserving semantic integrity, *FAST* effectively minimizes the gradient variance caused by arbitrary text concatenation present in baseline methods. This advantage is robust and sustained: the final column demonstrates that *FAST* maintains the highest GSNR in nearly $100\%$ of the steps. These findings provide strong empirical evidence that our method induces more stable gradients, which theoretically accounts for its superior generalization performance.

### 5.4. Ablation on Context Length

A potential confounding variable in out primary evaluation is the difference in maximum context length between *FAST*(truncation at 8192 tokens) and the Block Training baselines (fixed context size of 2048 tokens). To isolate the structural effect of sequential processing from the absolute context length, we conduct an ablation study using the Llama-3.1-8B-Instruct (Layer 20, 8X, JumpReLU) configuration.

We train an ablated *FAST* model where the sequence truncation is strictly set to 2,048 tokens (denoted *FAST-2048*), perfectly matching the context window size of the *BT(F)* baseline. The reconstruction fidelity and behavioral preservation metrics are presented in Table 2

*FAST-2048* performs almost identically to the default *FAST-8192*, while still drastically outperforming the *BT(F)* baseline at the exact same 2048-token limit (e.g., reducing Overall KL divergence by approximately 3x). This proves that the performance leap of *FAST* is fundamentally driven by respecting instance boundaries to eliminate cross-context pollution, rather than merely expanding context window.

### 5.5. Controlled-prefix Experiment

To empirically isolate the impact of unrelated context on training stability and validate our theoretical analysis of attention contamination, we conduct a controlled-prefix experiment. Using the *FAST* method as a clean baseline, we introduce an ablation variant denoted as RandomConcat. For each data, we prepend a random 512-token sequence sampled from an unrelated corpus. This design intentionally reintroduces the cross-instance attention leakage term ($\epsilon_{poll}$) into the gradient, while holding all other *FAST* parameters constant.

As detailed in Table 17, reintroducing this unrelated history leads to a sustained decrease in GSNR across the critical early training phase (Steps 0-100). The significant fluctuations and overall lower GSNR in RandomConcat empirically confirm that unrelated context acts as destructive noise rather than beneficial regularization for instruct models. Because the presence of an unrelated prefix is the sole isolated variable, this resulting GSNR drop provides direct empirical evidence that preserving data continuity is the fundamental driver of the optimization stability and training improvements observed in *FAST*.

## 6. Feature Interpretability

To rigorously assess feature quality, we employ a two-stage evaluation framework: first verifying numerical stability to filter noise, and then evaluating semantic interpretability via automated scoring.

**Feature Activation Stability.** Prior to semantic evaluation, we validate numerical stability to distinguish genuine signals from noise (details in Appendix L.1). Our analysis reveals a stark contrast: while *BT* baselines frequently exhibit cluttered, noisy activation patterns near zero, *FAST* demonstrates a clean separation between inactive and active states. This superior stability ensures that subsequent evaluations are grounded on robust, trustworthy features.

**Semantic Interpretability.** Building on stable features, we assess semantic coherence using an automated framework (Bills et al., 2023; He et al., 2024), where GPT-4o scores features based on their top-5 activating contexts. To quantitatively assess the quality of the learned features, we utilize a rigorous scoring rubric. As defined in Table 3, this 5-point scale measures monosemanticity, ranging from completely uninterpretable patterns (Score 1) to highly consistent, monosemantic features (Score 5). To ensure reproducibility, we detail the specific experimental setup for the SAEs. Table 20 outlines the target layers and expansion factors chosen for the Llama and Qwen model families. We target middle-to-late layers where semantic features are typically richest. *FAST* consistently outper-

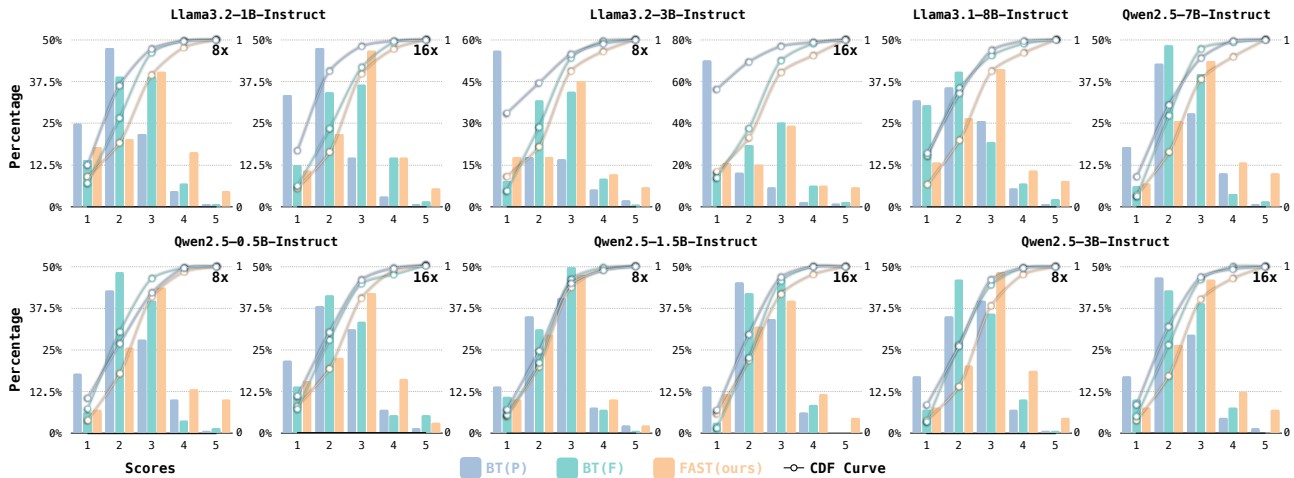

*Figure 6.* Feature interpretability scores distribution. *FAST* demonstrates superior interpretability across all models, achieving 21.1% high-quality features (scores 4–5) on Llama3.2-3B-Instruct, compared to 7.0% for *BT(P)* and 10.2% for *BT(F)*.

*Table 3.* Scoring criteria for feature interpretability.

| Score | Description |
|---|---|
| 5 | Clear pattern with no deviating examples |
| 4 | Clear pattern with one or two deviating examples |
| 3 | Clear overall pattern but quite a few examples not fitting that pattern |
| 2 | Broad consistent theme but lacking structure |
| 1 | No discernible pattern |

forms baselines as shown in Figure 6. Specifically, for Llama3.2-3B-Instruct, *FAST* yields 21.1% high-quality features (scores 4–5), significantly surpassing *BT(P)* (7.0%) and *BT(F)* (10.2%). This trend is reinforced by CDF analysis on Qwen2.5-3B-Instruct, where *FAST* exhibits the lowest proportion of low-quality features ($\leq 3$) at 76.5%, compared to over 89% for baselines. These results confirm that *FAST*'s sequence-based training effectively suppresses noise and enhances semantic clarity.

## 7. Steering with SAE Latents

Feature steering evaluates model inference by adjusting activation coefficients within a trained SAE. Following SAE reconstruction, we implement steering by scaling a selected latent dimension $k$ with coefficient $\alpha$:

$$z' = z + \alpha d_k \qquad (19)$$

where $d_k$ is the decoder vector and $z'$ is introduced into the model's residual stream space via the SAE decoder $W_{dec}$, and the resulting reconstruction is then integrated into

the model's forward pass. This approach enables precise control over specific semantic components. We conduct experiments using 1,010 instruction instances with scaling $\alpha \in [0, 15, 25, 50, 100, 150, 200]$ on 8X JumpReLU SAE through *FAST*, focusing on features related to special tokens (e.g., `<|start_header_id|>`, `<|im_start|>`). We focus on special tokens due to their fundamental role in structuring conversation.

Results show that steering high-activation features associated with special tokens significantly influences model output quality and reasoning ability. We observe a clear inverted-U relationship: moderate amplification ($\alpha = 15$-$50$) improves engagement and relevance, while excessive amplification ($\alpha > 100$) degrades output quality through repetitive responses. This suggests these features encode essential reasoning capabilities, presenting a "sweet spot" for feature steering that enhances performance without introducing biases. Detailed results are provided in Appendix M.4. Building upon this discovery, such initial explorations underscore a promising pathway for future research to harness SAE latents for dynamic, task-specific model optimization.

## 8. Conclusion

This paper introduces *FAST*, a novel paradigm for training SAEs on instruct models. By preserving the semantic integrity of individual instances, *FAST* effectively mitigates the context fragmentation and misalignment inherent in *BT*. Extensive experiments demonstrate that *FAST* consistently outperforms baselines, achieving higher GSNR and superior feature interpretability. Furthermore, we demonstrate the practical utility of *FAST*-trained SAEs by showing that feature steering on special tokens significantly enhances generation quality.

## Acknowledgements

Min Yang is supported by National Key Research and Development Program of China (2024YFF0908200), National Natural Science Foundation of China (Grant No. 62376262), and the Natural Science Foundation of Guangdong Province of China (2024A1515030166, 2025B1515020032).

## Impact Statement

This paper presents *FAST*, a novel training framework designed to enhance the interpretability of instruct models by aligning the training process with the sequential nature of fine-tuning data, effectively addressing the distributional mismatch and gradient noise inherent in Block Training. *FAST* contributes to the field of mechanistic interpretability by producing higher-fidelity reconstructions and semantically consistent features, offering new pathways for ensuring AI safety and alignment. *FAST* utilizes widely recognized, publicly available datasets for training and evaluation, strictly adhering to their licenses and usage policies without intentionally introducing private, personally identifiable information (PII) or offensive content.

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

# A. Additional Related Work

In this section, we provide an extended review of the background on mechanistic interpretability and the theoretical underpinnings of Sparse Autoencoders.

**Mechanistic Interpretability Challenges.** As LLMs continue to advance, their increasing complexity, massive parameter scales, and intricate training processes present significant challenges to human understanding (Bubeck et al., 2023; Bengio et al., 2024). Achieving a deep understanding is crucial to ensuring alignment with human values (Ji et al., 2025) and mitigating intended outcomes (Hendrycks et al., 2021). Mechanistic interpretability seeks to achieve a detailed understanding of model behavior through systematic reverse engineering, addressing the obscurity of underlying misalignment causes (Casper et al., 2024).

**Theoretical Foundations: Linear Representation and Superposition.** The training of SAEs is framed as dictionary learning, where hidden layer weights serve as the basis and sparsity constraints enforce efficient representations (Bereska & Gavves, 2024). SAEs align with the *linear representation hypothesis*, which suggests that features correspond to directions in activation space (Mikolov et al., 2013). This enables embedding arithmetic, such as $v(\text{"king"}) - v(\text{"man"}) + v(\text{"woman"}) = v(\text{"queen"})$.

Furthermore, SAEs address the phenomenon described by the *superposition hypothesis* (Elhage et al., 2022; Olah et al., 2020). Since neurons in LLMs are often polysemantic (encoding multiple features due to limited dimensionality), superposition explains how networks represent more features than available neurons by encoding them as nearly orthogonal directions. Although non-orthogonal overlaps introduce interference, the benefits of representing a greater number of features outweigh the drawbacks in high-dimensional spaces (Bricken et al., 2023; Rajamanoharan et al., 2024a). These theoretical properties make SAEs particularly valuable for decomposing language models into understandable components (Gao et al., 2024; Lieberum et al., 2024).

# B. Extended Theoretical Analysis of Optimization Dynamics and Efficiency Bounds

In this section, we quantify the optimization penalty suggested by the GSNR degradation in Theorem 3.4. We analyze the training dynamics of SAEs as a non-convex stochastic optimization problem. While our experiments use Adam, we derive variance-dependent convergence upper bounds under Stochastic Gradient Descent (SGD) as a tractable proxy for understanding how gradient noise affects optimization efficiency.

### B.1. Formal Setup and Assumptions

Let $\theta \in \mathbb{R}^d$ denote the SAE parameters, and let

$$\mathcal{L}(\theta) = \mathbb{E}_{x \sim \mathcal{D}} \left[ \ell_{\text{rec}}(x, \theta) + \lambda R(z_\theta(x)) \right]$$

denote the expected surrogate objective, where $z_\theta(x)$ is the SAE latent activation and $R(\cdot)$ denotes a differentiable surrogate of the sparsity regularizer. This smoothed objective is used only for the theoretical analysis; the argument is intended to capture the variance-dependent optimization effect rather than provide a convergence guarantee for the exact non-smooth SAE training objective. In practice, this surrogate is used to model the sparsity-inducing terms appearing in Standard SAE and JumpReLU SAE training. We analyze the update rule $\theta_{t+1} = \theta_t - \eta g(\theta_t)$, where $\eta$ is a constant learning rate and $g(\theta_t)$ is the stochastic gradient estimator.

We adopt standard assumptions for non-convex analysis (Ghadimi & Lan, 2013):

**Assumption B.1** ($L$-Smoothness). The surrogate objective $\mathcal{L}$ has $L$-Lipschitz continuous gradients. This implies the quadratic upper bound for any $\theta, \theta'$:

$$\mathcal{L}(\theta') \leq \mathcal{L}(\theta) + \langle \nabla \mathcal{L}(\theta), \theta' - \theta \rangle + \frac{L}{2} \|\theta' - \theta\|_2^2. \tag{20}$$

**Assumption B.2** (Unbiased Gradient and Bounded Variance). Let $\mathcal{F}_t$ be the filtration generated by the iterates up to time $t$. The stochastic gradient satisfies

$$\mathbb{E}[g(\theta_t) \mid \mathcal{F}_t] = \nabla \mathcal{L}(\theta_t), \tag{21}$$

and its conditional variance is bounded by a constant $\sigma^2$:

$$\mathbb{E}\left[\|g(\theta_t) - \nabla\mathcal{L}(\theta_t)\|_2^2 \mid \mathcal{F}_t\right] \leq \sigma^2. \tag{22}$$

Here, for a vector-valued stochastic gradient, we use the total vector variance

$$\mathrm{Var}(g(\theta_t) \mid \mathcal{F}_t) := \mathbb{E}\left[\|g(\theta_t) - \mathbb{E}[g(\theta_t) \mid \mathcal{F}_t]\|_2^2 \mid \mathcal{F}_t\right]. \tag{23}$$

Therefore,

$$\mathbb{E}[\|g(\theta_t)\|_2^2 \mid \mathcal{F}_t] = \|\nabla\mathcal{L}(\theta_t)\|_2^2 + \mathrm{Var}(g(\theta_t) \mid \mathcal{F}_t) \leq \|\nabla\mathcal{L}(\theta_t)\|_2^2 + \sigma^2. \tag{24}$$

For Block Training, under Assumption 3.2, Theorem 3.4 connects the variance term in Assumption B.2 to attention contamination:

$$\sigma_{\mathrm{BT}}^2 \approx \sigma_{\mathrm{signal}}^2 + \sigma_{\mathrm{poll}}^2. \tag{25}$$

This connection allows the standard SGD bound below to quantify how the additional contamination variance affects optimization efficiency.

## B.2. Convergence Rate Analysis

We derive the convergence rate to an expected $\epsilon$-stationary point, i.e.,

$$\min_{0 \leq t < T} \mathbb{E}\left[\|\nabla\mathcal{L}(\theta_t)\|_2^2\right] \leq \epsilon.$$

**Theorem B.3** (Non-Convex SGD Convergence Bound). *Under Assumptions B.1 and B.2, suppose that $\mathcal{L}$ is lower bounded by $\mathcal{L}^*$. Then SGD with constant learning rate $\eta \leq \frac{1}{L}$ satisfies*

$$\min_{0 \leq t < T} \mathbb{E}\left[\|\nabla\mathcal{L}(\theta_t)\|_2^2\right] \leq \frac{2(\mathcal{L}(\theta_0) - \mathcal{L}^*)}{T\eta} + L\eta\sigma^2. \tag{26}$$

*Proof.* For notational simplicity, let $\mathcal{L}_t := \mathcal{L}(\theta_t)$, $\nabla\mathcal{L}_t := \nabla\mathcal{L}(\theta_t)$, and $g_t := g(\theta_t)$. By the $L$-smoothness of $\mathcal{L}$ and the SGD update $\theta_{t+1} = \theta_t - \eta g_t$, we have

$$\mathcal{L}_{t+1} \leq \mathcal{L}_t - \eta\langle\nabla\mathcal{L}_t, g_t\rangle + \frac{L\eta^2}{2}\|g_t\|_2^2. \tag{27}$$

Taking conditional expectation with respect to $\mathcal{F}_t$ and using Assumption B.2, we obtain

$$\begin{aligned}
\mathbb{E}[\mathcal{L}_{t+1} \mid \mathcal{F}_t] &\leq \mathcal{L}_t - \eta\|\nabla\mathcal{L}_t\|_2^2 + \frac{L\eta^2}{2}\mathbb{E}[\|g_t\|_2^2 \mid \mathcal{F}_t] \\
&\leq \mathcal{L}_t - \eta\|\nabla\mathcal{L}_t\|_2^2 + \frac{L\eta^2}{2}\left(\|\nabla\mathcal{L}_t\|_2^2 + \sigma^2\right) \\
&= \mathcal{L}_t - \left(\eta - \frac{L\eta^2}{2}\right)\|\nabla\mathcal{L}_t\|_2^2 + \frac{L\eta^2}{2}\sigma^2.
\end{aligned} \tag{28}$$

Since $\eta \leq 1/L$, we have $\eta - \frac{L\eta^2}{2} \geq \frac{\eta}{2}$. Therefore,

$$\frac{\eta}{2}\|\nabla\mathcal{L}_t\|_2^2 \leq \mathcal{L}_t - \mathbb{E}[\mathcal{L}_{t+1} \mid \mathcal{F}_t] + \frac{L\eta^2}{2}\sigma^2. \tag{29}$$

Taking total expectation and summing over $t = 0, \ldots, T-1$ gives

$$\frac{\eta}{2}\sum_{t=0}^{T-1}\mathbb{E}\left[\|\nabla\mathcal{L}_t\|_2^2\right] \leq \sum_{t=0}^{T-1}\left(\mathbb{E}[\mathcal{L}_t] - \mathbb{E}[\mathcal{L}_{t+1}]\right) + \frac{TL\eta^2}{2}\sigma^2. \tag{30}$$

The first term on the right-hand side telescopes to $\mathbb{E}[\mathcal{L}_0] - \mathbb{E}[\mathcal{L}_T]$. Since $\mathcal{L}$ is lower bounded by $\mathcal{L}^*$, we have $\mathbb{E}[\mathcal{L}_T] \geq \mathcal{L}^*$. Hence,

$$\frac{\eta}{2}\sum_{t=0}^{T-1}\mathbb{E}\left[\|\nabla\mathcal{L}_t\|_2^2\right] \leq \mathcal{L}(\theta_0) - \mathcal{L}^* + \frac{TL\eta^2}{2}\sigma^2. \tag{31}$$

Multiplying both sides by $\frac{2}{T\eta}$ and using $\min_t a_t \leq \frac{1}{T}\sum_{t=0}^{T-1} a_t$, we obtain

$$\min_{0 \leq t < T} \mathbb{E}\left[\|\nabla\mathcal{L}(\theta_t)\|_2^2\right] \leq \frac{2(\mathcal{L}(\theta_0) - \mathcal{L}^*)}{T\eta} + L\eta\sigma^2. \tag{32}$$

This completes the proof. □

### B.3. Efficiency Ratio and Adaptive Optimizers

Eq. (26) shows that the convergence bound contains two competing terms: an optimization term $\mathcal{O}(1/(T\eta))$, which decreases with the learning rate $\eta$, and a variance-dependent term $\mathcal{O}(\eta\sigma^2)$, which increases with $\eta$. Therefore, the learning rate needs to balance these two terms.

Let $\Delta = \mathcal{L}(\theta_0) - \mathcal{L}^*$. The bound in Eq. (26) can be written as

$$F(\eta) = \frac{2\Delta}{T\eta} + L\eta\sigma^2. \tag{33}$$

Ignoring the constraint $\eta \leq 1/L$, the minimizer satisfies

$$\eta^* = \sqrt{\frac{2\Delta}{LT\sigma^2}} = \mathcal{O}\left(\frac{1}{\sigma\sqrt{T}}\right). \tag{34}$$

When this unconstrained value violates the smoothness constraint, the learning rate can be clipped as $\eta = \min\{1/L, \eta^*\}$. Substituting the unconstrained optimizer into Eq. (26) gives

$$\min_{0 \leq t < T} \mathbb{E}\left[\|\nabla\mathcal{L}(\theta_t)\|_2^2\right] \leq 2\sqrt{\frac{2L\Delta\sigma^2}{T}} = \mathcal{O}\left(\frac{\sigma}{\sqrt{T}}\right). \tag{35}$$

Thus, to make this upper bound no larger than a squared-gradient tolerance $\epsilon$, it suffices to take

$$T \geq \frac{8L\Delta\sigma^2}{\epsilon^2}. \tag{36}$$

Equivalently, the iteration complexity suggested by this upper bound scales as

$$T_{\text{bound}}(\epsilon) = \mathcal{O}\left(\frac{\sigma^2}{\epsilon^2}\right). \tag{37}$$

**Corollary B.4** (Efficiency Penalty of Block Training). *Suppose that Block Training and FAST have comparable smoothness constants, initial optimality gaps, and target squared-gradient tolerance. Then the SGD-style bound-level iteration proxies scale proportionally to their gradient variances:*

$$\rho_{\text{bound}} := \frac{T_{\text{BT}}}{T_{\text{FAST}}} \approx \frac{\sigma_{\text{BT}}^2}{\sigma_{\text{FAST}}^2}. \tag{38}$$

*Using the variance decomposition $\sigma_{\text{BT}}^2 \approx \sigma_{\text{signal}}^2 + \sigma_{\text{poll}}^2$ and assuming FAST removes the modeled cross-instance contamination term, so that $\sigma_{\text{FAST}}^2 \approx \sigma_{\text{signal}}^2$, we obtain*

$$\rho_{\text{bound}} \approx 1 + \frac{\sigma_{\text{poll}}^2}{\sigma_{\text{signal}}^2}. \tag{39}$$

This bound-level analysis suggests that $\sigma_{\text{poll}}^2$ acts as a multiplicative penalty on the variance-dependent component of training efficiency.

*Remark* B.5 (Connection to Adam). Although the above derivation is for SGD, the same variance effect is consistent with the behavior of adaptive optimizers such as Adam. This remark is qualitative and is not intended as a convergence proof for Adam. Adam rescales each coordinate by

$$\frac{\hat{m}_{t,i}}{\sqrt{\hat{v}_{t,i}} + \epsilon_{\text{adam}}}, \tag{40}$$

where $\hat{m}_{t,i}$ and $\hat{v}_{t,i}$ are the first- and second-moment estimates for coordinate $i$. Under the decomposition $g_{\text{BT},i} = g_{\text{signal},i} + \epsilon_{\text{poll},i}$, and assuming the perturbation is approximately zero-mean and uncorrelated with the signal, we have

$$\mathbb{E}[g_{\text{BT},i}^2] \approx \mathbb{E}[g_{\text{signal},i}^2] + \mathbb{E}[\epsilon_{\text{poll},i}^2]. \tag{41}$$

Thus, attention-contamination noise can inflate the coordinate-wise second-moment estimate $\hat{v}_{t,i}$, which may reduce the effective adaptive step size in noisy coordinates. This observation is consistent with the variance-dependent penalty suggested by Eq. (37).

## C. SAE Initialization Method

The encoder weights ($W_{enc}$) and decoder weights ($W_{dec}$) are initialized using the Kaiming Uniform initialization method (He et al., 2015). This step, used exclusively in the JumpReLU method, normalizes each row of the $W_{dec}$ using the L2 norm and adjusts the threshold $\epsilon$ and encoder bias $b_{enc}$ accordingly. After that, some data is selected for geometric median evaluation. The goal is to minimize the weighted sum of distances to all sample points. To achieve this, the Weiszfeld algorithm is employed to a specified precision of $\text{ftol} = 1 \times 10^{-20}$. The resulting optimal point is then used as the initial value for $b_{\text{dec}}$. There exists the formulas about the geometric median evaluation as follows:

$$f(\mathbf{m}) = \sum_{i=1}^{n} w_i \|\mathbf{m} - \mathbf{p}_i\|, \mathbf{m}_0 = \frac{\sum_{i=1}^{n} w_i \mathbf{p}_i}{\sum_{i=1}^{n} w_i} \tag{42}$$

$$d_i = \|\mathbf{p}_i - \mathbf{m}_k\|, w_i' = \frac{w_i}{\max(d_i, \epsilon)} \tag{43}$$

$$\mathbf{m}_{k+1} = \frac{\sum_{i=1}^{n} w_i' \mathbf{p}_i}{\sum_{i=1}^{n} w_i'} \tag{44}$$

$$|f(\mathbf{m}_{k+1}) - f(\mathbf{m}_k)| \leq \text{ftol} \cdot f(\mathbf{m}_k) \tag{45}$$

The parameters used in the equations are defined as follows: $\mathbf{m}$ represents the target point or the weighted mean to be optimized, while $\mathbf{p}_i$ is the $i$-th data point in the dataset. $w_i$ denotes the weight associated with the $i$-th data point. The objective function, $f(\mathbf{m})$, is the weighted sum of distances between $\mathbf{m}$ and all data points $\mathbf{p}_i$. The initial estimate of $\mathbf{m}$, denoted as $\mathbf{m}_0$, is calculated as the weighted mean of all points. $d_i$ is the distance between the $i$-th data point $\mathbf{p}_i$ and the current estimate $\mathbf{m}_k$. The updated weight for the $i$-th data point, $w_i'$, is adjusted by the distance $d_i$ and a small constant $\epsilon$ to prevent division by zero. $\mathbf{m}_{k+1}$ is the updated estimate of $\mathbf{m}$ at iteration $k+1$, computed as the weighted mean of all points using the updated weights $w_i'$.

## D. SFT Dataset Construction Details

We collect and integrate several large-scale instruction datasets specifically designed for fine-tuning LLMs. Datasets are shown below:

- **WildChat-1M-Full** (Zhao et al., 2024) is a dataset comprising 1 million conversations between human users and ChatGPT, enriched with demographic metadata such as state, country, hashed IP addresses, and request headers.

- **Infinity-Instruct** (Li et al., 2025) is a large-scale, high-quality instruction dataset, specifically designed to enhance the instruction-following capabilities of LLMs in both general and domain-specific tasks.

- **tulu-3-sft-mixture** (Lambert et al., 2025) is used to train the Tulu 3 series of models

- **orca-agentinstruct-1M-v1-cleaned** [4] is a cleaned version of the orca-agentinstruct-1M-v1 (Mitra et al., 2024) dataset released by Microsoft, a fully synthetic dataset using only raw text publicly available on the web as seed data.

---

[4]https://huggingface.co/datasets/mlabonne/orca-agentinstruct-1M-v1-cleaned

*Table 4.* Length distribution of training instances in FAST. The vast majority (96.6%) of instances contain fewer than 2048 tokens, indicating that using `context_size`=2048 for BT provides comparable effective context coverage in most cases.

| Length Range | Count | Percentage |
|---|---|---|
| $\leq$ 2048 tokens | $N_{\leq 2048}$ | 96.6% |
| > 2048 tokens | $N_{>2048}$ | 3.4% |
| Total | $N_{\text{total}}$ | 100% |

- **lmsys-chat-1m** (Zheng et al., 2023) is a comprehensive real-world conversational dataset containing one million interactions with 25 LLMs. This dataset spans a wide range of topics and interaction types, effectively capturing diverse user-LLM interaction patterns.

Together, they comprise 11,425,231 samples, forming a robust and diverse foundation for advancing research on instruct LLMs. Inevitably, many datasets contain a significant amount of similar or even duplicate data, which can adversely affect both model training and the accuracy of evaluations. To address this issue, we employ an n-gram-based deduplication technique to preprocess the data (Algorithm 1). N-gram method decomposes text into consecutive sequences of n words (or characters), effectively capturing local features.

This approach enables the detection and identification of repetitive patterns within the text. By leveraging this method, we are able to filter out not only completely identical instances but also content that exhibits high semantic or structural similarity. Consequently, the quality and diversity of the dataset are significantly enhanced. Finally, we adopt a 20-gram deduplication strategy to eliminate redundancy in the dataset. After applying this process, a total of 4,758,226 data entries are obtained.

---

**Algorithm 1** Deduplicate Dataset by N-Grams

---

**Input:** Dataset $\mathcal{D}$, N-gram size $n$
**Output:** Deduplicated dataset $\mathcal{D}_{dedup}$
  $\mathcal{D}_{dedup} \leftarrow \{\}$ # Initialize deduplicated dataset
  $seen\_hashes \leftarrow \{\}$ # Set to store hashes of seen N-grams
  **for** each sample $s$ in $\mathcal{D}$ **do**
    $ngrams \leftarrow \{\}$ # Initialize N-grams for the sample
    **for** each conversation $c$ in $s.conversations$ **do**
      $ngrams \leftarrow ngrams \cup \text{GenerateNGrams}(c.content, n)$
    **end for**
    **if** any $\text{Hash}(ngram) \in seen\_hashes$ for $ngram \in ngrams$ **then**
      **continue** #Skip sample if any N-gram hash is already seen
    **end if**
    $seen\_hashes \leftarrow seen\_hashes \cup \{\textbf{Hash}(ngram) \mid ngram \in ngrams\}$
    $\mathcal{D}_{dedup} \leftarrow \mathcal{D}_{dedup} \cup \{s\}$
  **end for**
  **return** $\mathcal{D}_{dedup}$

---

## E. SAE Model Implementation Details

This section details the mathematical formulations of the two SAE architectures employed in *FAST*.

**Standard ReLU SAE.** The Standard SAE utilizes a ReLU activation function. The encoder, decoder, and loss function are defined as follows:

$$f(\mathbf{x}) = \text{ReLU}(\mathbf{W}^{\text{enc}}\mathbf{x} + \mathbf{b}^{\text{enc}}) \tag{46}$$

$$\hat{\mathbf{x}} = \mathbf{W}^{\text{dec}}f(\mathbf{x}) + \mathbf{b}^{\text{dec}} \tag{47}$$

$$\mathcal{L} = \|\mathbf{x} - \hat{\mathbf{x}}\|_2^2 + \lambda\|\mathbf{z}\|_1 \tag{48}$$

Here, $\mathbf{W}^{\text{enc}}, \mathbf{W}^{\text{dec}}, \mathbf{b}^{\text{enc}}, \mathbf{b}^{\text{dec}}$ denote the learnable weights and biases. The term $\|\mathbf{z}\|_1$ represents the $L_1$ norm of the feature activations used for sparsity regularization, scaled by the hyperparameter $\lambda$.

**JumpReLU SAE.** The JumpReLU SAE modifies the activation function to allow for separate control over the activation threshold and magnitude. The formulation is:

$$f(\mathbf{x}) = \text{JumpReLU}_\theta(\mathbf{W}^{\text{enc}}\mathbf{x} + \mathbf{b}^{\text{enc}}) \tag{49}$$

$$\hat{\mathbf{x}} = \mathbf{W}^{\text{dec}}f(\mathbf{x}) + \mathbf{b}^{\text{dec}} \tag{50}$$

$$\mathcal{L} = \|\mathbf{x} - \hat{\mathbf{x}}\|_2^2 + \lambda\|\mathbf{z}\|_0 \tag{51}$$

The activation function is defined as $\text{JumpReLU}_\theta(z) := z \odot H(z - \theta)$, where $\theta > 0$ is a learnable threshold parameter, $\odot$ denotes element-wise multiplication, and $H$ is the Heaviside step function. The sparsity term $\|\mathbf{z}\|_0$ approximates the $L_0$ norm (count of non-zero elements).

## F. Hyperparameter Settings

The detailed parameter settings used in the experiment are as follows:

*Table 5.* Layer configurations of the Llama and Qwen model series, showcasing the selection of layers across varying depths to mitigate depth-related biases and optimize model performance.

| Model Name | Layer |
|---|---|
| *Llama series* | |
| Llama-3.1-8B-Instruct | [4,12,18,20,25] |
| Llama-3.2-3B-Instruct | [4,12,20] |
| Llama-3.2-1B-Instruct | [4,9,14] |
| *Qwen series* | |
| Qwen2.5-7B-Instruct | [4,12,18,20,25] |
| Qwen2.5-3B-Instruct | [4,18,32] |
| Qwen2.5-1.5B-Instruct | [4,14,24] |
| Qwen2.5-0.5B-Instruct | [4,12,20] |

**General Settings**

- Learning Rate ($lr$): $7 \times 10^{-5}$

- End Learning Rate ($lr_{end}$): $7 \times 10^{-6}$

- Seed: $42$

- Data Type ($dtype$): `float32`

**Optimizer Settings**

- Optimizer: Adam
  - Beta 1 ($\beta_1$): 0.9
  - Beta 2 ($\beta_2$): 0.999

- Learning Rate Scheduler: `cosineannealing`
  - Learning Rate Decay Steps: $64,000$
  - Learning Rate Warm-up Steps: $16,000$

- Sparsity Loss Coefficient ($L_{\text{sparsity}}$):
  - 0.01 for JumpReLU
  - 5 for Standard

- Sparsity Loss Warm-up Steps ($L_{\text{sparsity}}$): $10,000$

**Training Settings**

- Training Tokens: $4.096 \times 10^7$

- Train Batch Size (tokens): $128$

- Mixing Activation Buffer Size: $262,144$
  - Note: Calculated as 32 contexts $\times$ $8,192$ tokens, following the default configuration of `sae_lens`.

**Activation and Decoder Initialization**

- Decoder Initialization Method ($b_{dec\_init\_method}$): `geometric_median`

- Normalize SAE Decoder: `True`

- Dead Feature Threshold: $10^{-8}$

- Dead Feature Window: 1000

**Additional Settings**

- Noise Scale: 0

- Expansion Factor: 8 or 16

- Feature Sampling Window: 2000

- JumpReLU Bandwidth: 0.001

- JumpReLU Init Threshold: 0.001

- Apply Decoder to Input ($apply\_b\_dec\_to\_input$): `False`

- Use Ghost Gradients: `False`

- Use Cached Activations: `False`

# G. Mean Squared Error (MSE) of SAEs

The Mean Squared Error (MSE) results for the token reconstruction task are presented in this section.

## G.1. Mean Squared Error (MSE) of special tokens of standard SAEs

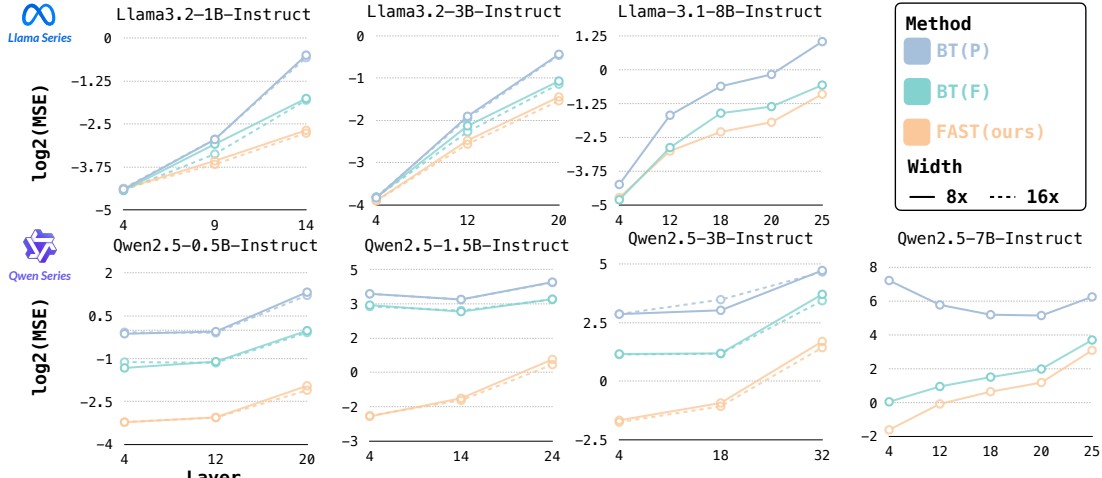

*Figure 7.* $\text{MSE}_{st}$ performance of the Standard SAE (all metrics are presented in log scale, where lower values indicate better SAE reconstruction performance). Within the Standard architecture, *FAST* exhibits the best reconstruction capability compared to *BT(P)* and *BT(F)*

While the reconstruction capability of Standard SAE models was generally inferior to the JumpReLU structure, *FAST* is also able to effectively reduce the $\text{MSE}_{st}$, especially in the Qwen series models.

## G.2. MSE of SAEs trained on Llama-3.1-8B-Instruct

*Table 6.* Mean Squared Error (MSE) of SAEs trained on Llama-3.1-8B-Instruct. Each value is highlighted with a green background to indicate performance, with darker shades of green representing better results.

| Layer | Expansion Factor | Method | Standard SAE | | JumpReLU SAE | |
|---|---|---|---|---|---|---|
| | | | $\log_2(\text{MSE})$ | $\log_2(\text{MSE}_{st})$ | $\log_2(\text{MSE})$ | $\log_2(\text{MSE}_{st})$ |
| 4 | 8 | *BT(P)* | -5.5059 | -4.2377 | -9.4350 | -6.8026 |
| | | *BT(F)* | -5.6080 | -4.8046 | -9.8097 | -8.3853 |
| | | *FAST* | -5.6432 | -4.7236 | -9.8187 | -10.1534 |
| 12 | 8 | *BT(P)* | -3.2837 | -1.6776 | -11.2353 | -5.4823 |
| | | *BT(F)* | -3.3437 | -2.8733 | -13.9975 | -9.2049 |
| | | *FAST* | -3.4104 | -3.0011 | -14.1393 | -12.1287 |
| 18 | 8 | *BT(P)* | -1.6059 | -0.6085 | -13.0282 | -7.4267 |
| | | *BT(F)* | -1.7131 | -1.6009 | -15.0851 | -10.4278 |
| | | *FAST* | -1.8697 | -2.2923 | -15.0666 | -12.4442 |
| 20 | 8 | *BT(P)* | -1.1852 | -0.1692 | -13.3080 | -7.8271 |
| | | *BT(F)* | -1.3509 | -1.3587 | -14.7969 | -10.4507 |
| | | *FAST* | -1.4721 | -1.9375 | -15.5552 | -13.1463 |
| 25 | 8 | *BT(P)* | -0.1677 | 1.0444 | -12.9767 | -7.1657 |
| | | *BT(F)* | -0.5163 | -0.5639 | -16.6192 | -11.6569 |
| | | *FAST* | -0.5747 | -0.8982 | -16.5138 | -15.9845 |

## G.3. MSE of SAEs trained on Llama-3.2-3B-Instruct

*Table 7.* Mean Squared Error (MSE) of SAEs trained on Llama-3.2-3B-Instruct. Each value is highlighted with a green background to indicate performance, with darker shades of green representing better results.

| Layer | Expansion Factor | Method | Standard SAE | | JumpReLU SAE | |
|---|---|---|---|---|---|---|
| | | | $\log_2(\mathrm{MSE})$ | $\log_2(\mathrm{MSE}_{st})$ | $\log_2(\mathrm{MSE})$ | $\log_2(\mathrm{MSE}_{st})$ |
| 4 | 8 | BT(P) | -4.5650 | -3.8363 | -13.7434 | -8.3908 |
| | | BT(F) | -4.5785 | -3.8250 | -13.6105 | -8.5868 |
| | | FAST | -4.5931 | -3.9053 | -9.0852 | -8.7193 |
| | 16 | BT(P) | -4.5645 | -3.8158 | -9.6278 | -7.5321 |
| | | BT(F) | -4.5858 | -3.8210 | -9.6102 | -7.6905 |
| | | FAST | -4.5959 | -3.9055 | -9.8054 | -9.3065 |
| 12 | 8 | BT(P) | -2.6239 | -1.9052 | -13.4038 | -8.5246 |
| | | BT(F) | -2.6757 | -2.1318 | -14.7879 | -9.1440 |
| | | FAST | -2.7236 | -2.4763 | -15.3747 | -13.4614 |
| | 16 | BT(P) | -2.6279 | -1.9488 | -12.2827 | -7.7836 |
| | | BT(F) | -2.6754 | -2.2725 | -13.8874 | -8.4299 |
| | | FAST | -2.7509 | -2.5644 | -14.4420 | -12.6355 |
| 20 | 8 | BT(P) | -0.6926 | -0.4378 | -13.5554 | -8.4006 |
| | | BT(F) | -0.9186 | -1.0709 | -14.8424 | -8.9061 |
| | | FAST | -0.9527 | -1.4473 | -18.8809 | -17.3707 |
| | 16 | BT(P) | -0.8145 | -0.4607 | -13.1516 | -9.1137 |
| | | BT(F) | -1.0947 | -1.1447 | -14.2900 | -8.9611 |
| | | FAST | -1.1285 | -1.5387 | -14.6872 | -12.1711 |

## G.4. MSE of SAEs trained on Llama-3.2-1B-Instruct

*Table 8.* Mean Squared Error (MSE) of SAEs trained on Llama-3.2-1B-Instruct. Each value is highlighted with a green background to indicate performance, with darker shades of green representing better results.

| Layer | Expansion Factor | Method | Standard SAE | | JumpReLU SAE | |
|---|---|---|---|---|---|---|
| | | | $\log_2(\mathrm{MSE})$ | $\log_2(\mathrm{MSE}_{st})$ | $\log_2(\mathrm{MSE})$ | $\log_2(\mathrm{MSE}_{st})$ |
| 4 | 8 | BT(P) | -5.3374 | -4.4021 | -15.3160 | -9.6296 |
| | | BT(F) | -5.3583 | -4.4375 | -15.6237 | -10.0324 |
| | | FAST | -5.3775 | -4.3920 | -15.8654 | -13.9127 |
| | 16 | BT(P) | -5.3370 | -4.3794 | -14.5574 | -9.0583 |
| | | BT(F) | -5.3587 | -4.4358 | -14.7275 | -9.4817 |
| | | FAST | -5.3804 | -4.3879 | -10.5009 | -10.2448 |
| 9 | 8 | BT(P) | -3.6638 | -2.9507 | -7.9900 | -7.2577 |
| | | BT(F) | -3.7759 | -3.0874 | -16.1021 | -10.5349 |
| | | FAST | -3.8282 | -3.5754 | -16.4928 | -13.9685 |
| | 16 | BT(P) | -3.6642 | -2.9456 | -7.1584 | -6.5155 |
| | | BT(F) | -3.8049 | -3.3775 | -15.1966 | -9.8149 |
| | | FAST | -3.8344 | -3.6778 | -15.8696 | -12.9629 |
| 14 | 8 | BT(P) | -1.2195 | -0.4927 | -8.0419 | -5.1825 |
| | | BT(F) | -1.7311 | -1.7559 | -15.2996 | -9.3409 |
| | | FAST | -1.7410 | -2.6844 | -21.4449 | -23.4395 |
| | 16 | BT(P) | -1.2449 | -0.5642 | -6.4784 | -5.2817 |
| | | BT(F) | -1.8371 | -1.8036 | -14.9445 | -9.3654 |
| | | FAST | -1.8409 | -2.7668 | -16.2748 | -13.3547 |

## G.5. MSE of SAEs trained on Qwen2.5-7B-Instruct

*Table 9.* Mean Squared Error (MSE) of SAEs trained on Qwen2.5-7B-Instruct. Each value is highlighted with a green background to indicate performance, with darker shades of green representing better results.

| Layer | Expansion Factor | Method | Standard SAE | | JumpReLU SAE | |
|---|---|---|---|---|---|---|
| | | | $\log_2(\text{MSE})$ | $\log_2(\text{MSE}_{st})$ | $\log_2(\text{MSE})$ | $\log_2(\text{MSE}_{st})$ |
| 4 | 8 | BT(P) | 1.2919 | 7.2207 | -4.1852 | 1.9109 |
| | | BT(F) | -0.5233 | 0.0494 | -5.9622 | -3.3368 |
| | | FAST | -0.7358 | -1.6090 | -10.6174 | -11.9105 |
| 12 | 8 | BT(P) | 1.4751 | 5.7788 | -5.8014 | -4.1171 |
| | | BT(F) | 0.7681 | 0.9550 | -6.3039 | -5.9309 |
| | | FAST | 0.6177 | -0.0770 | -9.8207 | -10.4545 |
| 18 | 8 | BT(P) | 2.0024 | 5.1985 | -6.5926 | -4.0005 |
| | | BT(F) | 1.4749 | 1.5093 | -6.8466 | -8.0743 |
| | | FAST | 1.3892 | 0.6468 | -9.1659 | -9.7604 |
| 20 | 8 | BT(P) | 2.6772 | 5.1501 | -4.9649 | -0.7776 |
| | | BT(F) | 2.1453 | 1.9877 | -5.6461 | -3.5904 |
| | | FAST | 2.0796 | 1.1869 | -8.2213 | -8.7821 |
| 25 | 8 | BT(P) | 4.8764 | 6.2532 | -2.1482 | 2.0938 |
| | | BT(F) | 4.4139 | 3.7031 | -2.6957 | 1.6207 |
| | | FAST | 4.4471 | 3.0934 | -4.9598 | -5.5615 |

## G.6. MSE of SAEs trained on Qwen2.5-3B-Instruct

*Table 10.* Mean Squared Error (MSE) of SAEs trained on Qwen2.5-3B-Instruct. Each value is highlighted with a green background to indicate performance, with darker shades of green representing better results.

| Layer | Expansion Factor | Method | Standard SAE | | JumpReLU SAE | |
|---|---|---|---|---|---|---|
| | | | $\log_2(\text{MSE})$ | $\log_2(\text{MSE}_{st})$ | $\log_2(\text{MSE})$ | $\log_2(\text{MSE}_{st})$ |
| 4 | 8 | BT(P) | -0.8873 | 2.8616 | -8.7177 | -2.2147 |
| | | BT(F) | -1.4572 | 1.1595 | -8.5340 | -1.9954 |
| | | FAST | -1.5098 | -1.6682 | -13.9907 | -11.6534 |
| | 16 | BT(P) | -1.0058 | 2.8627 | -8.8511 | -2.3755 |
| | | BT(F) | -1.6685 | 1.1371 | -8.9769 | -2.4576 |
| | | FAST | -1.5147 | -1.7482 | -13.2162 | -10.7660 |
| 18 | 8 | BT(P) | 0.9257 | 3.0243 | -9.2313 | -2.9916 |
| | | BT(F) | 0.4744 | 1.1862 | -9.3796 | -2.9188 |
| | | FAST | 0.6782 | -0.9288 | -10.3007 | -11.2916 |
| | 16 | BT(P) | 0.8594 | 3.4799 | -9.6147 | -3.1930 |
| | | BT(F) | 0.3438 | 1.1729 | -9.5534 | -3.0426 |
| | | FAST | 0.5485 | -1.0730 | -10.3197 | -11.1114 |
| 32 | 8 | BT(P) | 3.8883 | 4.7227 | -4.3442 | -2.3480 |
| | | BT(F) | 3.4388 | 3.7056 | -5.5300 | -5.3856 |
| | | FAST | 3.6647 | 1.6953 | -5.0278 | -7.3022 |
| | 16 | BT(P) | 3.7736 | 4.6584 | -4.4299 | -2.9327 |
| | | BT(F) | 3.2978 | 3.4334 | -5.6515 | -6.2729 |
| | | FAST | 3.5676 | 1.4331 | -5.0783 | -7.2653 |

### G.6.1. QWEN2.5-1.5B-INSTRUCT

*Table 11.* Mean Squared Error (MSE) of SAEs trained on Qwen2.5-1.5B-Instruct. Each value is highlighted with a green background to indicate performance, with darker shades of green representing better results.

| Layer | Expansion Factor | Method | Standard SAE | | JumpReLU SAE | |
|---|---|---|---|---|---|---|
| | | | $\log_2(\mathrm{MSE})$ | $\log_2(\mathrm{MSE}_{st})$ | $\log_2(\mathrm{MSE})$ | $\log_2(\mathrm{MSE}_{st})$ |
| 4 | 8 | BT(P) | -0.1150 | 3.8222 | -5.0404 | 1.5111 |
| | | BT(F) | -0.5653 | 3.2719 | -5.1794 | 1.3737 |
| | | FAST | -0.7745 | -2.1358 | -13.4069 | -12.5193 |
| | 16 | BT(P) | -0.2315 | 3.8196 | -4.8980 | 1.6550 |
| | | BT(F) | -0.7614 | 3.2068 | -5.1495 | 1.4045 |
| | | FAST | -0.9958 | -2.0996 | -13.3622 | -11.6841 |
| 14 | 8 | BT(P) | 0.4087 | 3.5463 | -5.4990 | 1.0522 |
| | | BT(F) | 0.0306 | 2.9569 | -6.2791 | 0.2762 |
| | | FAST | -0.0925 | -1.2535 | -11.2579 | -11.8198 |
| | 16 | BT(P) | 0.3186 | 3.5454 | -4.9561 | 1.5981 |
| | | BT(F) | -0.0918 | 3.0073 | -5.9567 | 0.5989 |
| | | FAST | -0.2312 | -1.3543 | -11.6309 | -12.1911 |
| 24 | 8 | BT(P) | 3.0506 | 4.3907 | -4.6425 | 0.4759 |
| | | BT(F) | 2.5424 | 3.5608 | -5.3630 | 0.5141 |
| | | FAST | 2.5122 | 0.6336 | -6.2603 | -7.9484 |
| | 16 | BT(P) | 2.9411 | 4.3725 | -4.4566 | 1.1218 |
| | | BT(F) | 2.3877 | 3.5499 | -5.0298 | 1.0916 |
| | | FAST | 2.3762 | 0.3794 | -6.3063 | -8.0686 |

## G.7. MSE of SAEs trained on Qwen2.5-0.5B-Instruct

*Table 12.* Mean Squared Error (MSE) of SAEs trained on Qwen2.5-0.5B-Instruct. The best and second-best methods are highlighted with dark green and light green backgrounds, respectively.

| Layer | Expansion Factor | Method | Standard SAE | | JumpReLU SAE | |
|---|---|---|---|---|---|---|
| | | | $\log_2(\mathrm{MSE})$ | $\log_2(\mathrm{MSE}_{st})$ | $\log_2(\mathrm{MSE})$ | $\log_2(\mathrm{MSE}_{st})$ |
| 4 | 8 | BT(P) | -2.7554 | -0.1257 | -10.6725 | -4.1202 |
| | | BT(F) | -2.8808 | -1.3213 | -11.6763 | -5.1212 |
| | | FAST | -2.8732 | -3.2218 | -21.7343 | -23.1697 |
| | 16 | BT(P) | -2.9204 | -0.0721 | -10.7024 | -4.1569 |
| | | BT(F) | -3.1034 | -1.1148 | -11.6959 | -5.1497 |
| | | FAST | -3.0970 | -3.2153 | -17.4590 | -16.7389 |
| 12 | 8 | BT(P) | -2.0463 | -0.0492 | -9.5392 | -2.9978 |
| | | BT(F) | -2.2811 | -1.1008 | -10.4276 | -3.8743 |
| | | FAST | -2.2836 | -3.0505 | -21.1734 | -25.6605 |
| | 16 | BT(P) | -2.1648 | -0.0915 | -9.4019 | -2.8551 |
| | | BT(F) | -2.4489 | -1.1418 | -10.5582 | -4.0043 |
| | | FAST | -2.4406 | -3.0602 | -20.7499 | -19.0931 |
| 20 | 8 | BT(P) | 0.2408 | 1.3303 | -10.5099 | -4.2017 |
| | | BT(F) | -0.3029 | -0.0174 | -11.4078 | -4.8666 |
| | | FAST | -0.3387 | -1.9461 | -15.2442 | -16.9599 |
| | 16 | BT(P) | 0.1296 | 1.2181 | -10.6728 | -4.2739 |
| | | BT(F) | -0.4536 | -0.0825 | -11.3337 | -4.7864 |
| | | FAST | -0.4924 | -2.1033 | -16.3662 | -18.0564 |

## G.8. MSE of SAEs trained on Qwen2.5-Math-7B

To address concerns regarding the generalizability of our method beyond standard instruction-tuned models, we conducted additional experiments on Qwen2.5-Math-7B using JumpReLU SAEs. We evaluated the performance on representative layers (12, 18, 20, and 25).

As shown in Table 13, *FAST* consistently achieves significantly lower Log2(MSE) compared to baseline methods across all tested layers. For instance, in Layer 12, *FAST* reaches a Log2(MSE) of 0.09, showing a substantial improvement over *BT(F)* (0.78) and *BT(P)* (4.88). These results confirm that *FAST* maintains its robustness and effectiveness even when applied to specialized, domain-specific models.

*Table 13.* Log2(MSE) comparison on Qwen2.5-Math-7B.

| Model | Layer | *BT(P)* | *BT(F)* | *FAST* |
|---|---|---|---|---|
| | 12 | 4.88 | 0.78 | **0.09** |
| | 18 | 5.10 | 1.23 | **0.43** |
| Qwen2.5-Math-7B | 20 | 5.52 | 2.11 | **1.23** |
| | 25 | 5.93 | 2.85 | **1.77** |

# H. Delta Loss Comparison Across Different Models and Training Methods.

*Table 14.* Delta Loss Comparison Across Different Models and Training Methods.

| Model | Layer | Exp. | *BT(P)* | *BT(F)* | *FAST* |
|---|---|---|---|---|---|
| Llama-3.1-8B-it | 20 | 8X | +4.67% | -2.03% | **-0.51%** |
| Llama-3.2-1B-it | 9 | 16X | +19.57% | -0.60% | **-0.10%** |
| | | 8X | +11.50% | -0.15% | **-0.01%** |
| Llama-3.2-3B-it | 12 | 16X | +2.31% | +1.60% | **+0.02%** |
| | | 8X | +2.16% | +0.78% | **+0.04%** |
| Qwen2.5-0.5B-it | 12 | 16X | +8.47% | +7.47% | **-0.01%** |
| | | 8X | -1.16% | +2.20% | **+0.01%** |
| Qwen2.5-1.5B-it | 14 | 16X | +11.96% | +9.97% | **-0.06%** |
| | | 8X | +8.80% | +10.06% | **+0.37%** |
| Qwen2.5-3B-it | 18 | 16X | +51.43% | +39.86% | **-0.04%** |
| | | 8X | +46.25% | +41.19% | **-0.15%** |
| Qwen2.5-7B-it | 20 | 8X | -2.17% | +0.36% | **-0.04%** |

# I. KL Divergence Analysis of SAEs

The KL Divergence results for the token reconstruction task are presented in this section. We conduct this inference-only experiment on the JumpReLU architecture across all reported models, with Table 15 detailing the overall KL divergence across the entire vocabulary and Table 16 focusing on the reconstruction performance of special tokens. The results strongly corroborate the superiority of our *FAST* method, which achieves the lowest KL divergence in 11 out of 12 configurations. Notably, on Qwen2.5 models (0.5B and 3B), *FAST* reduces the divergence by orders of magnitude compared to Block Training baselines, indicating significantly better preservation of the original model's predictive distribution.

*Table 15.* Overall KL Divergence (Lower is Better). Comparison across different models and widths. The best, second-best, and third-best methods are highlighted with dark green, medium green, and light green backgrounds, respectively.

| Model | Layer | Width | KL Divergence (Lower is Better) | | |
|---|---|---|---|---|---|
| | | | *BT(P)* | *BT(F)* | *FAST*(Ours) |
| Qwen2.5-0.5B-Instruct | 12 | 16X | 2.05e-03 | 6.99e-04 | 2.30e-06 |
| | | 8X | 1.99e-03 | 9.46e-04 | 1.83e-06 |
| Qwen2.5-1.5B-Instruct | 14 | 16X | 1.82e-03 | 9.27e-04 | 1.71e-04 |
| | | 8X | 8.85e-04 | 5.47e-04 | 2.57e-04 |
| Qwen2.5-3B-Instruct | 18 | 16X | 6.29e-03 | 5.05e-03 | 3.32e-04 |
| | | 8X | 6.21e-03 | 4.42e-03 | 3.36e-04 |
| Qwen2.5-7B-Instruct | 20 | 8X | 3.67e-04 | 2.63e-04 | 4.38e-04 |
| Llama-3.2-1B-Instruct | 9 | 16X | 5.06e-02 | 2.84e-04 | 1.21e-04 |
| | | 8X | 3.11e-02 | 7.75e-05 | 7.40e-05 |
| Llama-3.2-3B-Instruct | 12 | 16X | 2.73e-04 | 8.80e-05 | 7.82e-05 |
| | | 8X | 1.21e-04 | 4.36e-05 | 3.24e-05 |
| Llama-3.1-8B-Instruct | 20 | 8X | 9.95e-04 | 8.19e-05 | 2.59e-05 |

*Table 16.* **Special Token KL Divergence (Lower is Better).** Highlighting the stability of control tokens. *FAST* demonstrates superior performance in most configurations.

| Model | Layer | Width | Special Token KL (Lower is Better) | | |
|---|---|---|---|---|---|
| | | | *BT(P)* | *BT(F)* | *FAST*(Ours) |
| Qwen2.5-0.5B-Instruct | 12 | 16X | 9.64e-03 | 3.38e-03 | 7.72e-06 |
| | | 8X | 5.77e-03 | 5.09e-03 | 3.11e-06 |
| Qwen2.5-1.5B-Instruct | 14 | 16X | 3.69e-03 | 1.90e-03 | 9.45e-04 |
| | | 8X | 3.97e-03 | 2.00e-03 | 1.45e-03 |
| Qwen2.5-3B-Instruct | 18 | 16X | 1.67e-01 | 1.11e-01 | 1.21e-03 |
| | | 8X | 1.53e-01 | 9.80e-02 | 1.26e-03 |
| Qwen2.5-7B-Instruct | 20 | 8X | 3.44e-04 | 2.93e-05 | 1.50e-03 |
| Llama-3.2-1B-Instruct | 9 | 16X | 4.01e-02 | 5.35e-03 | 1.40e-04 |
| | | 8X | 2.88e-02 | 7.47e-04 | 8.89e-05 |
| Llama-3.2-3B-Instruct | 12 | 16X | 7.36e-04 | 3.99e-04 | 3.39e-04 |
| | | 8X | 2.94e-04 | 2.08e-04 | 1.19e-04 |
| Llama-3.1-8B-Instruct | 20 | 8X | 2.93e-03 | 1.02e-03 | 5.69e-04 |

## J. GSNR Results in Controlled-prefix Ablation Study

*Table 17.* GSNR comparison in the controlled-prefix ablation study. Introducing a random unrelated prefix (Random-Prefix) consistently degrades the GSNR compared to the clean FAST baseline, validating the destructive nature of attention contamination.

| Model | Method | 0 | 10 | 20 | 30 | 40 | 50 | 60 | 70 | 80 | 90 | 100 |
|---|---|---|---|---|---|---|---|---|---|---|---|---|
| Qwen2.5-7B | FAST | 0.43 | 0.50 | 0.52 | 0.53 | 0.50 | 0.31 | 0.30 | 0.33 | 0.33 | 0.31 | 0.28 |
| | RandomConcat | 0.40 | 0.22 | 0.48 | 0.36 | 0.20 | 0.30 | 0.28 | 0.30 | 0.19 | 0.19 | 0.20 |
| Llama-3.1-8B | FAST | 0.50 | 0.45 | 0.41 | 0.45 | 0.46 | 0.40 | 0.39 | 0.37 | 0.36 | 0.36 | 0.37 |
| | RandomConcat | 0.28 | 0.24 | 0.26 | 0.24 | 0.25 | 0.23 | 0.25 | 0.25 | 0.24 | 0.21 | 0.20 |

# K. Training Efficiency Analysis

We address the potential concern regarding the computational overhead of our proposed method. We recorded the total training duration for SAEs on both Qwen2.5-7B-Instruct and Llama-3.1-8B-Instruct (Layer 20, Width 8X) under identical hardware conditions.

As detailed in Table 18, *FAST* introduces no additional computational overhead compared to Block Training (BT). On the contrary, empirical measurements demonstrate that *FAST* is consistently more efficient, reducing total training time by approximately 15% to 21% across different architectures.

This efficiency gain is attributed to the streamlined data pipeline of *FAST*. Unlike Block Training, which concatenates documents into fixed-length sequences (often necessitating padding or truncation logic during processing), *FAST* processes data instances sequentially and independently. Since the majority (96.6%) of instruction tuning data instances are shorter than the standard context window (2048 tokens), our sequential processing strategy proves to be computationally superior in practice, effectively eliminating the processing of padding tokens.

*Table 18.* Comparison of Training Efficiency. Total training time measured on Layer 20 with 8X expansion width. FAST consistently achieves significant time reduction compared to baselines across different models and SAE architectures.

| Model | Architecture | Method | Training Time | Reduction (vs *BT(P)*) |
|---|---|---|---|---|
| Qwen2.5-7B-Instruct | Standard SAE | *BT(P)* | 16h 31m 21s | – |
| | | *BT(F)* | 16h 12m 55s | – |
| | | *FAST* | **13h 40m 35s** | **~17.2%** |
| | JumpReLU SAE | *BT(P)* | 16h 22m 56s | – |
| | | *BT(F)* | 16h 12m 49s | – |
| | | *FAST* | **14h 00m 30s** | **~14.5%** |
| Llama-3.1-8B-Instruct | Standard SAE | *BT(P)* | 17h 40m 18s | – |
| | | *BT(F)* | 17h 43m 38s | – |
| | | *FAST* | **14h 09m 38s** | **~19.9%** |
| | JumpReLU SAE | *BT(P)* | 17h 44m 05s | – |
| | | *BT(F)* | 16h 56m 23s | – |
| | | *FAST* | **13h 56m 13s** | **~21.4%** |

# L. Implementation Details of Feature Interpretability

This section presents the detailed implementation aspects of our feature interpretability evaluation. We explicitly analyze the distribution of feature activations to categorize their effectiveness and provide a comprehensive overview of the model configurations, scoring criteria, and the specific prompts used in our automated evaluation pipeline.

### L.1. Analysis of Feature Activation Distribution

To distinguish between truly meaningful features and numerical noise, we move beyond the binary distinction of "dead" vs. "alive" features. Instead, we categorize feature activations into three distinct zones based on their maximum activation values ($v$) over a sampled dataset:

- **Weak/Dead Zone** ($v \leq 0.5$)**:** Features that rarely activate or remain dormant within the sampled context.

- **Effective Zone** ($0.5 < v \leq 10$)**:** Features in this range typically represent stable and interpretable concepts. A higher count here indicates superior semantic capture.

- **Unstable Zone** ($v > 10$)**:** Features with extremely high activations, often indicating numerical instability or "feature explosion" rather than precise semantic alignment.

We evaluated 10,000 samples on Qwen2.5-7B-Instruct and Llama-3.1-8B-Instruct. The distribution of maximum feature activation values is summarized in Table 19.

*Table 19.* Distribution of Maximum Feature Activation Values. *FAST* consistently activates a larger absolute number of features in the "Effective Zone." Notably, in Qwen2.5-7B, baselines suffer from significant feature explosion (>21% unstable features), which *FAST* effectively mitigates to just 1.0%.

| Model | Method | Total Features | Weak [0, 0.5] | Effective (0.5, 10] | Unstable (>10) |
|-------|--------|----------------|---------------|---------------------|----------------|
| Qwen2.5-7B-Instruct | *BT(P)* | | 7521 (26.2%) | 15108 (52.7%) | 6043 (21.1%) |
| | *BT(F)* | 28672 | 9685 (33.8%) | 14237 (49.7%) | 4750 (16.6%) |
| | *FAST* | | **6728 (23.4%)** | **21662 (75.6%)** | **282 (1.0%)** |
| Llama-3.1-8B-Instruct | *BT(P)* | | 19226 (58.7%) | 13542 (41.3%) | 0 (0.0%) |
| | *BT(F)* | 32768 | 21050 (64.2%) | 11717 (35.8%) | 1 (0.0%) |
| | *FAST* | | **17779 (54.3%)** | **14989 (45.7%)** | **0 (0.0%)** |

As shown in Table 19, *FAST* consistently activates a larger absolute number of features in the "Effective Zone." For Qwen2.5-7B, *FAST* yields 21,662 effective features, significantly outperforming *BT(P)* (15,108) and *BT(F)* (14,237). Similarly, for Llama-3.1-8B, *FAST* enhances the density of useful features compared to baselines.

Crucially, regarding feature stability, we observe model-specific behaviors. In the Qwen model, baseline methods suffer from severe "feature explosion", with over 21% of features falling into the Unstable Zone. *FAST* effectively mitigates this issue, reducing unstable features to just 1.0%. In contrast, while the Llama model is inherently more robust to block training artifacts, *FAST* maintains this stability while still delivering superior feature effectiveness.

### L.2. Model configurations of the Llama and Qwen model series in feature interpretability.

For reproducibility, Table 20 lists the target layers and expansion factors for the Llama and Qwen families, focusing on middle-to-late layers where semantic representations are most prominent.

*Table 20.* Model configurations of the Llama and Qwen model series in feature interpretability.

| Model | Layer | Exp. Factor |
|-------|-------|-------------|
| *Llama series* | | |
| Llama-3.1-8B-Instruct | 18 | 8X |
| Llama-3.2-3B-Instruct | 12 | 8X&16X |
| Llama-3.2-1B-Instruct | 9 | 8X&16X |
| *Qwen series* | | |
| Qwen2.5-7B-Instruct | 18 | 8X |
| Qwen2.5-3B-Instruct | 18 | 8X&16X |
| Qwen2.5-1.5B-Instruct | 14 | 8X&16X |
| Qwen2.5-0.5B-Instruct | 12 | 8X&16X |

## L.3. Prompt for Feature Interpretability

We employ an automated interpretation pipeline to scale the evaluation across thousands of features. The prompts provided below demonstrate how we instruct the evaluator model (e.g., GPT-4o) to analyze feature activations. The **System Prompt** sets the evaluation persona and strictly enforces the scoring rubric, while the **Prompt Template** formats the specific token activation contexts for the model to analyze.

---

**System Prompt**

We are analyzing the activation levels of features in a neural network. Each feature activates specific tokens in a text, and the activation value of each token indicates its relevance to the feature. Higher activation values signify a stronger association.

Your task is to evaluate the feature based on the following scoring rubric and assign it a monosemanticity score.

### Scoring Rubric: Activation Consistency

1: No discernible pattern

2: Broad consistent theme but lacking structure

3: Clear overall pattern but quite a few examples not fitting that pattern

4: Clear pattern with one or two deviating examples

5: Clear pattern with no deviating examples

### Instructions:

1. Analyze the context provided, which consists of a sequence of alternating tokens and their corresponding activation values.

2. Assign a score based on the activation consistency rubric.

3. Provide a descriptive name for the feature that captures its essence.

Example output: 'My final verdict score is: [[3]], feature name is [[Mathematical Problem Explanation]]'.

User: {prompt}

---

**Prompt Template**

Below is the context of feature {feature_index}, represented as sentences with tokens and their activation values:
{context}

---

# M. Implementation Details of Steering with SAE Latents

## M.1. Detailed Feature Steering Analysis

The proposed method conceptually resembles activation guidance using decoder latent vectors; however, the SAE framework offers a more robust and disentangled mechanism for control. These latent variables correspond to row vectors of $W_{\text{dec}}$, where a scalar $\alpha$ modulates the intensity of the $k$-th latent. Unlike conventional approaches that rely on direct, often opaque manipulation of hidden states, SAE-based feature steering provides interpretable control over specific semantic or structural components while maintaining the model's overall coherence.

To ensure robust evaluation, we randomly partitioned the 1,010 sampled instruction instances into two subsets: an identification set of 1,000 samples used to pinpoint highly activated SAE features, and an evaluation set of 10 samples to assess post-steering model outputs. Inference was performed using the specific chat templates corresponding to each instruct model. The 10 evaluation questions are listed in Figure 8. This design ensures statistical reliability in feature selection while providing sufficient diversity to capture the nuanced effects of steering across varying question complexities.

We specifically focus on features associated with special tokens[5] (see Table 21) to examine how these non-entity-specific tokens influence generation. To ensure reproducibility and eliminate selection bias, we adopted a deterministic selection criterion: for each target special token, we calculated the average maximum activation values across the 1,000-sample identification set and selected the single feature with the highest maximum activation value for steering demonstrations. Complete activation statistics are provided in this appendix. This choice targets special tokens due to their fundamental role in structuring conversational flow and contextual boundaries, making them ideal candidates for studying high-level model behavioral control.

We present three representative case studies to illustrate the steering effects. Due to space constraints, we primarily focus on the `<|start_header_id|>` feature for Llama-3.1-8B-Instruct and the `<|im_start|>` feature for Qwen2.5-7B-Instruct. Experiments employed an 8X JumpReLU SAE (trained via *FAST*) with scaling factors $\alpha \in [0, 15, 25, 50, 100, 150, 200]$. This range was calibrated to capture the full spectrum of effects, from subtle improvements at lower values to potential degradation at extreme amplification.

Results indicate a consistent pattern. For instance, in Question 3 (Figure 11 and Figure 12), amplifying the feature tied to `<|im_start|>` and `user` reveals a clear trajectory: moderate values of $\alpha$ significantly enhanced engagement and output relevance. However, excessive amplification ($\alpha > 100$) led to degradation, manifesting as language switching or incoherent repetition. This progression demonstrates an inverted-U relationship between steering intensity and output quality, suggesting that optimal feature activation exists within a bounded "sweet spot."

Similarly, in Question 4 (Figure 13 and Figure 14), steering the top-ranked feature associated with the `<|im_start|>` marker produced more convincing and logically structured responses within a specific coefficient range. Pushing $\alpha$ beyond this limit resulted in increased verbosity and circular reasoning. These patterns are mirrored in Question 2 (Figure 9 and Figure 10), reinforcing the generalizability of these findings across diverse domains.

The consistency of these results implies that these features likely encode essential "meta-cognitive" aspects of the model's reasoning process, transcending specific tasks. The existence of an optimal $\alpha$ range indicates that special token features act as high-level regulators—controlling coherence, relevance, and structural organization—rather than dictating specific content. This observation highlights the practical value of SAEs: steering structural features associated with special tokens emerges as a principled, reliable method to refine model guidance, enhancing response quality while preserving the natural flow and authenticity of generated content.

_______________

[5]Note that `user` and `assistant` roles are treated as special tokens, as they function structurally alongside markers like `<|im_start|>`.

**Question 1:**

How do I export constants and classes from a JavaScript module?

**Question 2:**

FINAL EXAM  Question 1. What was Elsie Marley profession?

**Question 3:**

lettre de mativation

**Question 4:**

请回答以下问题，找出铁锤和磁铁之间的主要区别是什么？

**Question 5:**

Summarize this article in one sentence.\n\nMedia playback is not supported on this device\nFarah pulled away from American Dathan Ritzenhein in the last mile in his first race since retaining his 5,000m and 10,000m Olympic titles in Rio.\nIn the women\"s race, Olympic 5,000m champion Vivian Cheruiyot of Kenya won in her first half marathon.\nScotland\"s Mark Telford took the men\"s wheelchair crown, a second ahead of fellow Briton Bret Crossley.\nThe Great North Run is the world\"s biggest half marathon and there were more than 41,000 runners taking part in this year\"s event from 178 nations.\nFarah, 33, was taken on a fast pace by former American 5,000m record holder Ritzenhein, but powered away with a mile to go and even had time to do a cheeky heel flip before he crossed the line in one hour and four seconds, the slowest of his three wins.\nBelgium\"s Emmanuel Bett, who ran the second half of the race almost on his own, crossed in third.\nFarah told BBC Sport: \"To be honest with you, I\"m knackered.\n\"I knew I had to work hard because Dathan is a former training partner and was running a great race.\n\"He put his foot down and tried to get rid of me because he knew I have amazing pace.\n\"It\"s good to finish the year on a high, what a year I\"ve had. I just want to go home now, chill out, see the kids, get up to no good. \"\n\nFind out about how to get into running with our special guide.\n\nThe women\"s race was billed as a shoot-out between middle distance greats Cheruiyot and three-time Olympic champion Tirunesh Dibaba.\nDibaba failed to keep pace with Cheruiyot and fellow Kenyan Priscah Jeptoo in the closing stages of the 13.1-mile course. It was Cheruiyot who took victory, producing a sprint finish to clock 1:07.54, just one second ahead of Jeptoo.\nCheruiyot said: \"I\"m so happy because it\"s my birthday. I found it tough with one kilometre to go but it\"s fantastic for me to end my season this way. \"\nMedia playback is not supported on this device\n\nSummary:

**Question 6:**

def intersection(list1, list2):\n    \"\"\"\n    This function returns a list of common elements between two lists: list1 and list2.\n\n    Parameters:\n    list1 (list): First list of elements\n    list2 (list): Second list of elements\n\n    Returns:\n    list: A list of common elements between list1 and list2\n    \"\"\"    # Complete the code to find the intersection of list1 and list2 using nested for loops\n    common_elements = []\n    for element1 in list1:\n        for element2 in list2:\n            if element1 == element2:\n                common_elements.append(element1)\n                break\n\n    return common_elements

**Question 7:**

Article:\n\nAuthor Carol Dunbar stands outside of her writing studio that is under her family's water tower on their property deep in the woods south of Superior, Wisconsin. Jed Carlson / Superior Telegram\n\nCarol Dunbar works at the desk inside her writing studio in her family's water tower in the woods south of Superior on Tuesday afternoon, Oct. 26, 2021. Jed Carlson / Superior Telegram\n\nCarol Dunbar speaks about her love of nature and living off the grid on her family's property deep in the woods south of Superior on Tuesday afternoon, Oct. 26, 2021. Jed Carlson / Superior Telegram\n\nCarol Dunbar looks through short writings and other trinkets that were her grandmothers that she keeps in her writing studio in her family's water tower in the woods south of Superior on Tuesday afternoon, Oct. 26, 2021. Jed Carlson / Superior Telegram\n\nAuthor Carol Dunbar talks about her struggles with the editing process of her upcoming novel as she sits in her writing studio in the family's water tower south of Superior on Tuesday afternoon, Oct. 26, 2021. Jed Carlson / Superior Telegram\n\nAuthor Carol Dunbar looks out of one of the windows of her writing studio in her family's water tower on their property deep in the woods south of Superior on Tuesday afternoon, Oct. 26, 2021. Jed Carlson / Superior Telegram\n\nSUPERIOR, Wis. — Carol Dunbar stepped through the woods as fallen leaves crunched beneath her feet. Her homestead south of Superior includes the main residence, her husband's workshop and a water tower. Living off the grid, the structure is a necessity for the homestead's water pressure — and for Dunbar's work.\n\n"Me getting into this water tower was finding a space where I could shut a door behind me to create," she said. "I wouldn't want any other kind of office, and it definitely has its challenges."\n\nThe novelist and freelance ghostwriter's computers, manuscripts and books all reside under what some might consider to be their worst enemy: "There are literally two 250-gallon tanks of water over my head right now," she said.\n\nYes, her office has flooded several times.\n\n"It's like being in a room that's pouring rain. It's awful, and I've had to make peace with that."\n\nTo see her work be so vulnerable makes it that much more endearing. "I know there's a really interesting metaphor about art and risk," she added.\n\nThere's no other space on their 80 acres where she can work the way she's able to here. After numerous floods and years spent working from the living room, her husband redid the space and built the staircase for better access and heat circulation.\n\nOriginally intended as a guest room, it's a 10-by-10 space on the second floor of the water tower. She calls it the cockpit.\n\nThere's a porch on the back and windows on all four sides, so "I feel like I'm writing in the treetops," she said.\n\nWhile she hears water moving through the pipes around her, "The view that it affords me and the peace that I have here in this little space, and it is little … I wouldn't trade it for anything."\n\nTHE SPACE\n\nLight floods in from every angle. Her sitting and standing desks, compliments of her husband, rest at the center and in a corner, an ancient-looking podium holds one of her numerous dictionaries; she likes to compare decades-old definitions to those of today.\n\nThere are several aloe plants, drawings on the wall, and a storyboard with pinned photos of a sculpture and an Irish skyline — inspiration for future works, she said.\n\nAn assortment of candles, one of which she lights daily before she begins. "It keeps me mindful that I'm trying to capture the best light, the best in human nature," she said.\n\nShe keeps a collection of notebooks, color-coated for whatever novel she's writing, in her office, in the car, by her bed, to help her document inspiration when it strikes. "I got very frustrated when I got a good idea or I'd hear a piece of dialogue or I'd finally know how to describe the snow on that day, and I would write it down and never find it again," she said.\n\nIt has helped, but she still has scraps of paper pinned to her notebook pages. "It's like leaving yourself love letters," she said, sorting through a pile.\n\nShe wrote her second novel in long-hand on paper. It's an accessible way to create away from a screen, she said.\n\nIn the corner rests a red cushioned chair that came from a Minneapolis alley. Around her desk she has taped quotes and reminders. "In the end, it all comes down to what we think we deserve," reads one.\n\nAlso a piece of wood with words: "You just have to trust your own madness — Clive Barker."\n\nDunbar cherishes a writing award and remnants of work kept on paper scraps, memorabilia from an ancestor who emigrated from Italy. While Dunbar's relative wasn't supported in pursuing writing, Dunbar feels her work today honors herself and her ancestor.\n\nHer book shelf holds works by Joyce Carol Oates, Jesmyn Ward, Barbara Kingsolver, and a treasured copy of Eleonora Duse's "The Mystic in the Theatre." Duse strove to eliminate...(Truncated)\n\nItalicize all instances of Carol Dunbar\"s quotes.

**Question 8:**

考虑由所有节肢动物组成的集合$B$，并让$C$是包含所有天牛属物种的$B$的子集。对于$C$中的每个$v$，我们定义一个函数$f(v)$，它描述了天牛独特的蜇刺机制。您的任务是提供不少于五段的全面概述天牛。在这样做时，请详细探讨它们的身体和行为特征以及它们在生态和进化适应中的适应性。特别是，我们要求您探讨它们鲜艳的色彩和密集的毛发如何作为防御机制抵御捕食者。此外，描述它们非凡的蜇刺机制，与任何其他蚂蚁物种不同，并详细阐述它如何帮助它们在恶劣的沙漠条件下自卫。此外，请深入探讨它们的进化历史，这使它们具备了令人难以置信的生存技能。最后，强调正在实施的保护天牛种群的持续保护措施，这些种群受到气候变化和栖息地破坏的不利影响。您的回答应该是广泛的、有理的和科学的，每一段都详细说明天牛生命周期各个方面之间的复杂相互关系。

**Question 9:**

What is the smallest prime factor of $600851475143$?

**Question 10:**

Develop a comprehensive branding strategy that includes a brand name, logo, tagline, packaging design, and marketing plan for a new line of organic, non-toxic, biodegradable cleaning products that are socially responsible and sustainably made. Ensure that the branding strategy effectively communicates the brand\"s unique selling proposition, target audience, brand personality, and brand voice through all touchpoints, including print and digital media, social media, in-store displays, and product demos. Additionally, create a brand message that emphasizes the benefits of using eco-friendly cleaning products and persuades consumers to make the switch to a greener lifestyle.

*Figure 8.* 10 questions content

## M.2. Special Tokens

*Table 21.* Tokens that control response generation and formatting in the Llama and Qwen model series.

| Token ID | Token |
|----------|-------|
| *Llama series* | |
| 882 | user |
| 78191 | assistant |
| 128006 | <\|start_header_id\|> |
| 128007 | <\|end_header_id\|> |
| 128009 | <\|eot_id\|> |
| *Qwen series* | |
| 872 | user |
| 77091 | assistant |
| 151644 | <\|im_start\|> |
| 151645 | <\|im_end\|> |

## M.3. Average Top 5 Max Activation Values and Their Corresponding Indices for Tokens across a 1000-Sample Dataset

*Table 22.* Top 5 Average Activation Values for Special Tokens in Llama3.1-8B-instruct with JumpReLU SAE

| Approach | Token | Top 5 Max Activation Value (Index:Value) | | | | |
|----------|-------|------|------|------|------|------|
| *BT(P)*[8X] | 882 | **4453**:0.8120 | **30511**:0.724 | **18547**:0.597 | **19110**:0.500 | **20505**:0.469 |
| | 78191 | **5188**:0.5030 | **1923**:0.4900 | **31873**:0.486 | **20505**:0.468 | **3187**:0.4620 |
| | 128006 | **2604**:7.1220 | **20523**:0.800 | **7428**:0.7330 | **24017**:0.702 | **16640**:0.678 |
| | 128007 | **23901**:1.193 | **7808**:0.5210 | **3268**:0.5180 | **20505**:0.477 | **30244**:0.473 |
| | 128009 | **20505**:0.744 | **25940**:0.653 | **7961**:0.6460 | **21317**:0.585 | **19110**:0.569 |
| *BT(F)*[8X] | 882 | **11765**:0.823 | **25025**:0.814 | **7043**:0.6880 | **16826**:0.562 | **21896**:0.560 |
| | 78191 | **30553**:0.536 | **9728**:0.5270 | **11435**:0.507 | **14565**:0.505 | **13234**:0.497 |
| | 128006 | **17784**:7.480 | **17355**:0.947 | **28634**:0.782 | **9333**:0.7710 | **27149**:0.744 |
| | 128007 | **23677**:1.002 | **6426**:0.6680 | **26136**:0.603 | **5783**:0.5720 | **26958**:0.526 |
| | 128009 | **23677**:0.834 | **7100**:0.7560 | **30568**:0.734 | **15188**:0.666 | **8346**:0.6430 |
| *FAST*[8X] | 882 | **22534**:0.611 | **13320**:0.470 | **29165**:0.464 | **19871**:0.428 | **29033**:0.418 |
| | 78191 | **16063**:0.463 | **13320**:0.461 | **19871**:0.460 | **32613**:0.441 | **22277**:0.399 |
| | 128006 | **22642**:4.392 | **2417**:0.7170 | **27839**:0.706 | **3095**:0.7030 | **10814**:0.654 |
| | 128007 | **30457**:2.489 | **19871**:0.532 | **6870**:0.4640 | **28096**:0.446 | **13266**:0.413 |
| | 128009 | **13822**:0.753 | **22277**:0.606 | **21866**:0.537 | **17489**:0.493 | **118**:0.41200 |

*Table 23.* Top 5 Average Activation Values for Special Tokens in Llama3.2-3B-instruct with JumpReLU SAE

| Approach | Token ID | Top 5 Max Activation Value (Index:Value) | | | | |
|---|---|---|---|---|---|---|
| *BT(P)*[8X] | 882 | **3817**:0.4550 | **11734**:0.430 | **505**:0.42200 | **23884**:0.417 | **14851**:0.380 |
| | 78191 | **6451**:0.3460 | **11061**:0.340 | **19811**:0.327 | **12369**:0.325 | **11734**:0.308 |
| | 128006 | **2064**:20.351 | **5699**:0.4090 | **14393**:0.399 | **7505**:0.3770 | **548**:0.37500 |
| | 128007 | **20232**:0.427 | **5095**:0.4000 | **19583**:0.393 | **23908**:0.362 | **3719**:0.3590 |
| | 128009 | **14536**:0.468 | **16718**:0.437 | **23736**:0.413 | **13925**:0.379 | **10211**:0.368 |
| *BT(P)*[16X] | 882 | **23287**:0.814 | **44336**:0.718 | **10727**:0.712 | **11701**:0.683 | **26467**:0.658 |
| | 78191 | **34602**:0.622 | **10655**:0.600 | **45414**:0.591 | **23156**:0.553 | **19333**:0.522 |
| | 128006 | **38076**:28.41 | **48766**:0.675 | **16639**:0.659 | **28134**:0.653 | **45**:0.621000 |
| | 128007 | **9822**:0.7530 | **39737**:0.659 | **5712**:0.6430 | **38496**:0.574 | **23156**:0.570 |
| | 128009 | **483**:0.79800 | **48233**:0.789 | **22660**:0.670 | **24339**:0.624 | **23774**:0.600 |
| *BT(F)*[8X] | 882 | **21524**:0.496 | **17981**:0.471 | **10125**:0.436 | **11210**:0.431 | **14456**:0.410 |
| | 78191 | **16126**:0.447 | **8704**:0.4470 | **20691**:0.418 | **19630**:0.393 | **10125**:0.365 |
| | 128006 | **15765**:21.39 | **1640**:0.5180 | **14456**:0.479 | **45**:0.459000 | **17981**:0.442 |
| | 128007 | **7814**:0.5120 | **24565**:0.489 | **1759**:0.4840 | **8704**:0.4390 | **14456**:0.396 |
| | 128009 | **5506**:0.5230 | **20691**:0.514 | **20328**:0.488 | **6878**:0.4550 | **7593**:0.4460 |
| *BT(F)*[16X] | 882 | **20561**:0.719 | **28995**:0.698 | **14625**:0.662 | **32041**:0.625 | **4844**:0.5850 |
| | 78191 | **23154**:0.725 | **8239**:0.6700 | **45582**:0.630 | **23594**:0.593 | **11425**:0.564 |
| | 128006 | **30984**:25.38 | **10207**:0.752 | **21441**:0.751 | **26876**:0.700 | **35477**:0.683 |
| | 128007 | **41219**:0.687 | **14625**:0.670 | **21050**:0.662 | **23942**:0.621 | **27267**:0.595 |
| | 128009 | **26876**:0.761 | **13612**:0.722 | **9537**:0.6930 | **44518**:0.653 | **6317**:0.6240 |
| *FAST*[8X] | 882 | **2950**:0.5730 | **1343**:0.5670 | **16808**:0.498 | **19508**:0.481 | **5931**:0.4590 |
| | 78191 | **23183**:0.548 | **263**:0.50900 | **8564**:0.4860 | **2680**:0.4750 | **23798**:0.472 |
| | 128006 | **8772**:37.471 | **20896**:0.610 | **2950**:0.6060 | **12126**:0.538 | **16622**:0.534 |
| | 128007 | **12955**:0.550 | **22995**:0.536 | **3339**:0.5080 | **7878**:0.4970 | **2950**:0.4730 |
| | 128009 | **7814**:0.5850 | **16940**:0.551 | **4605**:0.5080 | **12331**:0.493 | **4439**:0.4880 |
| *FAST*[16X] | 882 | **9447**:0.8380 | **5861**:0.7210 | **19741**:0.716 | **22320**:0.669 | **25160**:0.645 |
| | 78191 | **4177**:0.8220 | **43897**:0.719 | **18009**:0.667 | **25117**:0.594 | **30970**:0.590 |
| | 128006 | **22974**:37.66 | **36**:0.873000 | **18075**:0.813 | **26318**:0.774 | **45047**:0.762 |
| | 128007 | **42421**:0.798 | **655**:0.75300 | **13955**:0.697 | **26318**:0.632 | **28994**:0.589 |
| | 128009 | **29041**:0.888 | **18075**:0.844 | **33332**:0.776 | **2705**:0.7120 | **26318**:0.695 |

*Table 24.* Top 5 Average Activation Values for Special Tokens in Llama3.2-1B-instruct with JumpReLU SAE

| Approach | Token ID | Top 5 Max Activation Value (Index:Value) | | | | |
|---|---|---|---|---|---|---|
| *BT(P)*[8X] | 882 | **12248**:0.455 | **14322**:0.446 | **10030**:0.444 | **11886**:0.425 | **731**:0.39800 |
| | 78191 | **14903**:0.443 | **15672**:0.435 | **8014**:0.4190 | **13261**:0.410 | **11985**:0.405 |
| | 128006 | **4464**:10.463 | **4858**:0.4600 | **12143**:0.454 | **9898**:0.4440 | **6877**:0.3700 |
| | 128007 | **196**:0.45400 | **15332**:0.398 | **9561**:0.3580 | **12143**:0.355 | **626**:0.35500 |
| | 128009 | **15332**:0.496 | **1296**:0.4910 | **4858**:0.4170 | **6877**:0.4170 | **15975**:0.412 |
| *BT(P)*[16X] | 882 | **20612**:0.642 | **22827**:0.613 | **3012**:0.6050 | **11176**:0.578 | **2141**:0.5760 |
| | 78191 | **28423**:0.672 | **24765**:0.661 | **30621**:0.649 | **22827**:0.649 | **18585**:0.621 |
| | 128006 | **4169**:11.460 | **11176**:0.793 | **9495**:0.6770 | **9911**:0.6730 | **24072**:0.586 |
| | 128007 | **26090**:0.820 | **10861**:0.622 | **24072**:0.615 | **26939**:0.591 | **23109**:0.541 |
| | 128009 | **11176**:0.747 | **16525**:0.716 | **26594**:0.685 | **8403**:0.6490 | **15861**:0.633 |
| *BT(F)*[8X] | 882 | **2387**:0.4130 | **13266**:0.341 | **7778**:0.3090 | **8423**:0.2840 | **3682**:0.2800 |
| | 78191 | **7783**:0.3320 | **10427**:0.316 | **8941**:0.3150 | **16174**:0.311 | **4764**:0.3080 |
| | 128006 | **2537**:9.9460 | **15768**:0.382 | **9146**:0.3500 | **1604**:0.3440 | **14204**:0.312 |
| | 128007 | **10680**:0.390 | **15478**:0.312 | **8905**:0.3090 | **6638**:0.3020 | **15034**:0.284 |
| | 128009 | **2568**:0.4050 | **3528**:0.3860 | **14204**:0.371 | **1604**:0.3600 | **15768**:0.313 |
| *BT(F)*[16X] | 882 | **24100**:0.530 | **6794**:0.5240 | **7848**:0.5230 | **9322**:0.4900 | **17577**:0.490 |
| | 78191 | **12548**:0.583 | **24258**:0.542 | **2092**:0.5260 | **2460**:0.4960 | **15997**:0.484 |
| | 128006 | **4967**:10.559 | **24354**:0.675 | **20054**:0.614 | **12136**:0.599 | **12707**:0.537 |
| | 128007 | **18190**:0.581 | **2543**:0.5000 | **23285**:0.499 | **15997**:0.494 | **17059**:0.486 |
| | 128009 | **26830**:0.635 | **17228**:0.623 | **11407**:0.551 | **18494**:0.523 | **11681**:0.483 |
| *FAST*[8X] | 882 | **2926**:0.3780 | **878**:0.35400 | **4753**:0.3370 | **10237**:0.336 | **7582**:0.3140 |
| | 78191 | **13371**:0.388 | **14099**:0.376 | **8581**:0.3680 | **11313**:0.361 | **5121**:0.3400 |
| | 128006 | **12361**:8.486 | **13371**:0.386 | **878**:0.37500 | **129**:0.34900 | **1866**:0.3300 |
| | 128007 | **8581**:0.4120 | **12864**:0.357 | **13371**:0.341 | **4478**:0.3380 | **4523**:0.3150 |
| | 128009 | **878**:0.47000 | **11483**:0.408 | **6832**:0.3770 | **8581**:0.3690 | **865**:0.34700 |
| *FAST*[16X] | 882 | **1835**:0.7500 | **3851**:0.7100 | **982**:0.60400 | **9493**:0.6020 | **8463**:0.4780 |
| | 78191 | **19765**:0.596 | **14393**:0.539 | **28589**:0.512 | **2350**:0.4850 | **12592**:0.482 |
| | 128006 | **12329**:10.30 | **9838**:0.6440 | **13262**:0.592 | **1450**:0.5260 | **27818**:0.504 |
| | 128007 | **3368**:0.5820 | **31764**:0.568 | **16867**:0.518 | **16432**:0.503 | **9648**:0.4590 |
| | 128009 | **10365**:0.696 | **31406**:0.637 | **30028**:0.602 | **15515**:0.574 | **16339**:0.535 |

*Table 25.* Top 5 Average Activation Values for Special Tokens in Qwen2.5-7B-instruct with JumpReLU SAE

| Approach | Token ID | Top 5 Max Activation Value (Index:Value) | | | | |
|---|---|---|---|---|---|---|
| *BT(P)*[8X] | 872 | **12461**:9.058 | **439**:3.88000 | **19183**:2.978 | **18767**:2.889 | **13685**:1.992 |
| | 77091 | **2547**:2.9330 | **15678**:2.562 | **19183**:2.549 | **6508**:2.3290 | **4400**:2.0270 |
| | 151644 | **12461**:9.193 | **1261**:2.7050 | **6508**:2.3060 | **2547**:2.1240 | **4400**:2.1140 |
| | 151645 | **1261**:2.9730 | **2547**:2.8640 | **6508**:2.4140 | **18778**:2.223 | **13888**:2.118 |
| *BT(F)*[8X] | 872 | **4710**:6.3500 | **15390**:3.377 | **20684**:3.192 | **25558**:2.937 | **27629**:2.800 |
| | 77091 | **25558**:3.135 | **27629**:3.061 | **19040**:3.012 | **10759**:2.802 | **13257**:2.378 |
| | 151644 | **4710**:6.7170 | **10759**:3.412 | **11735**:3.049 | **28219**:2.749 | **26983**:2.596 |
| | 151645 | **28219**:3.130 | **11735**:2.692 | **2174**:2.4670 | **10614**:2.464 | **25812**:2.120 |
| *FAST*[8X] | 872 | **13794**:37.19 | **17783**:4.816 | **20022**:4.519 | **21950**:4.077 | **11739**:4.053 |
| | 77091 | **20022**:5.667 | **11739**:4.352 | **16782**:4.180 | **2670**:3.7810 | **13794**:3.731 |
| | 151644 | **13794**:39.87 | **20022**:5.418 | **7579**:4.1900 | **3817**:4.1890 | **26689**:4.023 |
| | 151645 | **20022**:4.463 | **2670**:3.6970 | **22845**:3.139 | **25469**:2.939 | **9676**:2.6890 |

*Table 26.* Top 5 Average Activation Values for Special Tokens in Qwen2.5-3B-instruct with JumpReLU SAE

| Approach | Token ID | Top 5 Max Activation Value (Index:Value) | | | | |
|---|---|---|---|---|---|---|
| *BT(P)*[8X] | 872 | **11485**:2.756 | **8925**:2.4490 | **3645**:2.4130 | **1600**:2.1160 | **2801**:2.0860 |
| | 77091 | **10992**:1.911 | **1600**:1.8300 | **15929**:1.777 | **14942**:1.747 | **12230**:1.677 |
| | 151644 | **7152**:132.52 | **2713**:2.0100 | **11354**:1.996 | **15302**:1.891 | **15795**:1.885 |
| | 151645 | **12297**:2.588 | **11352**:2.457 | **4096**:2.4520 | **10336**:2.429 | **10992**:2.214 |
| *BT(P)*[16X] | 872 | **14113**:2.010 | **12080**:1.750 | **18074**:1.739 | **14580**:1.720 | **2607**:1.4890 |
| | 77091 | **4047**:1.3860 | **27294**:1.294 | **3356**:1.2890 | **14113**:1.248 | **9469**:1.2420 |
| | 151644 | **32641**:150.0 | **14113**:1.362 | **7224**:1.3340 | **28068**:1.327 | **4741**:1.2860 |
| | 151645 | **23725**:1.696 | **14113**:1.674 | **25421**:1.669 | **68**:1.619000 | **9469**:1.5140 |
| *BT(F)*[8X] | 872 | **7603**:2.8380 | **3184**:2.7840 | **15060**:2.777 | **8391**:2.7390 | **6484**:2.3780 |
| | 77091 | **15060**:3.175 | **3373**:2.3530 | **7293**:2.3480 | **1317**:2.3398 | **7603**:2.2900 |
| | 151644 | **16236**:121.2 | **16225**:2.563 | **7603**:2.5000 | **7189**:2.4970 | **958**:2.43000 |
| | 151645 | **3104**:3.9910 | **1317**:3.4210 | **16225**:3.397 | **6700**:3.3500 | **15704**:3.101 |
| *BT(F)*[16X] | 872 | **23210**:2.320 | **29265**:1.807 | **11930**:1.767 | **28994**:1.712 | **2757**:1.5020 |
| | 77091 | **23210**:1.844 | **6805**:1.6570 | **20713**:1.564 | **11930**:1.544 | **29265**:1.483 |
| | 151644 | **31443**:153.4 | **23210**:2.160 | **5146**:2.0010 | **24831**:1.894 | **29265**:1.859 |
| | 151645 | **5146**:2.9880 | **5924**:2.4320 | **5572**:2.3420 | **12821**:2.078 | **24491**:1.502 |
| *FAST*[8X] | 872 | **2941**:3.4410 | **8775**:2.6400 | **10076**:2.625 | **12216**:2.178 | **776**:1.99600 |
| | 77091 | **2653**:3.6370 | **10076**:3.450 | **3411**:3.0540 | **9785**:2.5100 | **11618**:2.004 |
| | 151644 | **8775**:248.36 | **12291**:2.880 | **10076**:2.829 | **3411**:2.8280 | **13964**:2.566 |
| | 151645 | **10076**:4.538 | **12216**:3.775 | **12139**:3.729 | **4383**:3.5920 | **12209**:3.279 |
| *FAST*[16X] | 872 | **6863**:3.7600 | **9230**:2.9510 | **20605**:2.446 | **21312**:2.285 | **17408**:2.063 |
| | 77091 | **23681**:4.223 | **6863**:3.9440 | **17147**:3.059 | **10035**:2.969 | **4751**:2.7968 |
| | 151644 | **31443**:85.35 | **5599**:1.5974 | **9299**:1.5341 | **18964**:1.445 | **4751**:1.4220 |
| | 151645 | **23681**:3.000 | **6863**:2.4390 | **20511**:2.173 | **9230**:1.8215 | **17147**:1.517 |

*Table 27.* Top 5 Average Activation Values for Special Tokens in Qwen2.5-1.5B-instruct with JumpReLU SAE

| Approach | Token ID | Top 5 Max Activation Value (Index:Value) | | | | |
|---|---|---|---|---|---|---|
| *BT(P)*[8X] | 872 | **734**:312.441 | **2664**:2.5160 | **576**:2.31600 | **4162**:2.1050 | **9629**:2.1030 |
| | 77091 | **1656**:2.2670 | **3248**:2.2090 | **4162**:2.1040 | **4098**:2.0910 | **8997**:2.0460 |
| | 151644 | **734**:288.485 | **391**:1.92500 | **5536**:1.9240 | **11982**:1.660 | **11102**:1.625 |
| | 151645 | **11322**:1.905 | **734**:1.74500 | **1263**:1.6030 | **9637**:1.5900 | **12143**:1.499 |
| *BT(P)*[16X] | 872 | **15738**:261.8 | **3080**:1.4920 | **2724**:1.3730 | **19787**:1.372 | **17743**:1.258 |
| | 77091 | **2724**:1.3720 | **17351**:1.354 | **1954**:1.3340 | **19787**:1.307 | **9767**:1.2760 |
| | 151644 | **15738**:241.5 | **13157**:1.148 | **13486**:1.116 | **14339**:0.945 | **6977**:0.9250 |
| | 151645 | **9971**:1.1270 | **22929**:1.032 | **14028**:1.003 | **19840**:0.936 | **22072**:0.864 |
| *BT(F)*[8X] | 872 | **1910**:255.40 | **7039**:2.5590 | **9420**:2.5300 | **8118**:2.4710 | **1693**:2.4060 |
| | 77091 | **8118**:2.7040 | **7067**:2.5230 | **1223**:2.4890 | **7039**:2.4670 | **4086**:2.4190 |
| | 151644 | **1910**:234.85 | **4798**:1.9970 | **6153**:1.8900 | **5905**:1.7000 | **11021**:1.682 |
| | 151645 | **10536**:1.870 | **11021**:1.724 | **7064**:1.6550 | **1787**:1.5630 | **6153**:1.5040 |
| *BT(F)*[16X] | 872 | **2077**:263.49 | **13135**:1.624 | **17747**:1.439 | **16136**:1.353 | **19975**:1.338 |
| | 77091 | **6886**:1.5170 | **19975**:1.508 | **17747**:1.500 | **18492**:1.296 | **16136**:1.249 |
| | 151644 | **2077**:242.06 | **19387**:1.534 | **4177**:1.3580 | **22526**:1.283 | **19497**:1.178 |
| | 151645 | **4177**:1.1610 | **5724**:1.1000 | **9985**:1.0890 | **6552**:1.0190 | **11894**:0.945 |
| *FAST*[8X] | 872 | **7505**:462.49 | **4918**:2.4010 | **4694**:2.3060 | **4141**:2.1620 | **10728**:2.098 |
| | 77091 | **491**:2.25800 | **4141**:2.2300 | **11303**:2.125 | **8603**:2.0090 | **6358**:1.9430 |
| | 151644 | **7505**:425.73 | **10900**:1.793 | **6473**:1.7560 | **10139**:1.614 | **2006**:1.5990 |
| | 151645 | **491**:2.20100 | **11115**:1.748 | **11252**:1.665 | **6473**:1.5530 | **10257**:1.326 |
| *FAST*[16X] | 872 | **21852**:580.0 | **11515**:1.988 | **9360**:1.5720 | **21118**:1.501 | **11834**:1.487 |
| | 77091 | **21118**:2.068 | **9718**:1.6120 | **14362**:1.536 | **9360**:1.5240 | **11834**:1.477 |
| | 151644 | **21852**:532.9 | **21118**:1.683 | **16522**:1.350 | **17617**:1.265 | **12233**:1.174 |
| | 151645 | **21118**:2.070 | **17617**:1.474 | **16522**:1.312 | **18955**:1.196 | **21139**:1.084 |

*Table 28.* Top 5 Average Activation Values for Special Tokens in Qwen2.5-0.5B-insturct with JumpReLU SAE

| Approach | Token ID | Top 5 Max Activation Value (Index:Value) | | | | |
|---|---|---|---|---|---|---|
| *BT(P)*[8X] | 872 | **6091**:1.0680 | **2897**:0.8250 | **1389**:0.8240 | **6239**:0.8150 | **6434**:0.7770 |
| | 77091 | **3245**:0.8430 | **1767**:0.8430 | **1389**:0.8310 | **5981**:0.8120 | **6239**:0.7790 |
| | 151644 | **1608**:43.209 | **6818**:0.7600 | **6245**:0.7480 | **6724**:0.7150 | **1235**:0.7150 |
| | 151645 | **4541**:0.8170 | **5212**:0.8010 | **1744**:0.7760 | **4498**:0.7280 | **507**:0.72400 |
| *BT(P)*[16X] | 872 | **8475**:0.6880 | **13976**:0.545 | **889**:0.51000 | **8786**:0.4680 | **3099**:0.4680 |
| | 77091 | **3099**:0.5480 | **9308**:0.5340 | **13976**:0.528 | **8786**:0.4830 | **432**:0.46500 |
| | 151644 | **10161**:28.27 | **7726**:0.4830 | **6509**:0.4550 | **9343**:0.4510 | **6947**:0.4260 |
| | 151645 | **1934**:0.5580 | **12380**:0.505 | **7726**:0.4370 | **7385**:0.4370 | **1823**:0.4280 |
| *BT(F)*[8X] | 872 | **5375**:1.0290 | **3317**:0.9000 | **4825**:0.8510 | **3896**:0.8360 | **5791**:0.8260 |
| | 77091 | **3896**:0.8510 | **4825**:0.8450 | **2552**:0.8420 | **5375**:0.8030 | **3203**:0.8010 |
| | 151644 | **2428**:40.999 | **5130**:0.7510 | **1326**:0.7050 | **557**:0.68100 | **2765**:0.6540 |
| | 151645 | **2734**:0.8970 | **6507**:0.7080 | **628**:0.69600 | **2913**:0.6930 | **1119**:0.6680 |
| *BT(F)*[16X] | 872 | **13102**:0.658 | **12215**:0.572 | **10208**:0.542 | **6285**:0.4670 | **5598**:0.4430 |
| | 77091 | **7823**:0.5860 | **12215**:0.580 | **10208**:0.551 | **12606**:0.521 | **5598**:0.4871 |
| | 151644 | **1983**:27.761 | **5393**:0.5180 | **12215**:0.458 | **5515**:0.4470 | **9460**:0.4360 |
| | 151645 | **4484**:0.4980 | **12615**:0.472 | **13322**:0.441 | **5393**:0.4370 | **8592**:0.3820 |
| *FAST*[8X] | 872 | **1299**:0.9310 | **2747**:0.9090 | **3288**:0.8170 | **1859**:0.7860 | **4804**:0.7210 |
| | 77091 | **6296**:0.8960 | **6776**:0.8640 | **3288**:0.8450 | **7041**:0.8300 | **2747**:0.8140 |
| | 151644 | **3154**:34.650 | **825**:0.71700 | **5377**:0.6940 | **6140**:0.6830 | **3724**:0.6450 |
| | 151645 | **3724**:0.8630 | **3955**:0.8240 | **1371**:0.8030 | **3931**:0.6940 | **5940**:0.6740 |
| *FAST*[16X] | 872 | **11717**:0.578 | **6739**:0.5030 | **8487**:0.4990 | **2010**:0.4640 | **12647**:0.442 |
| | 77091 | **8487**:0.5840 | **6739**:0.5340 | **11717**:0.529 | **11505**:0.493 | **2851**:0.4760 |
| | 151644 | **3384**:28.324 | **4241**:0.4720 | **9335**:0.4250 | **11285**:0.416 | **298**:0.38400 |
| | 151645 | **4241**:0.5410 | **5731**:0.4450 | **6167**:0.4440 | **7780**:0.3940 | **5314**:0.3770 |

## M.4. Steering Output of Three Questions

### M.4.1. Q2

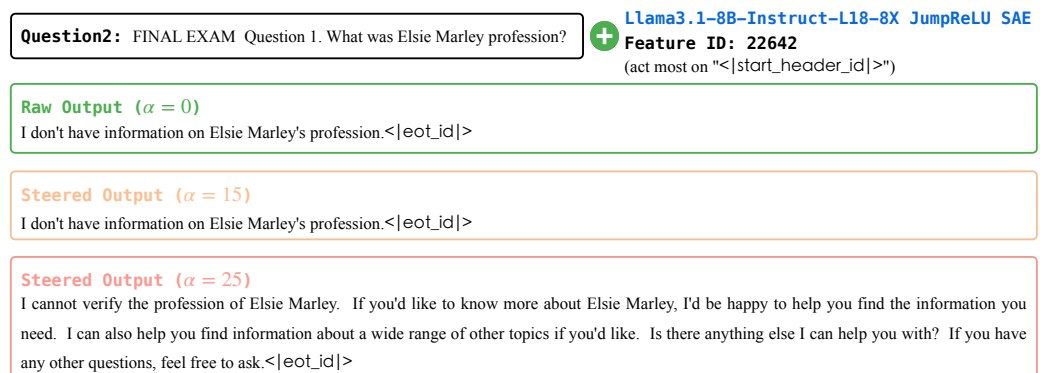

*Figure 9.* The steering output generated by Qwen2.5-7B-Instruct with Feature ID: 13794, focusing on `user` and `<|im_start|>` tokens for the Question 2 (entity description).

*Figure 10.* The steering output generated by Llama3.1-8B-Instruct with Feature ID: 22642, focusing on `user` and `<|im_start|>` tokens for the Question 2 (entity description).

For Question 2, the Qwen model (Figure 9) shows improved output quality when feature 13794 is moderately amplified ($\alpha$ = 25-75). Within this range, responses become more polite, detailed, and engaging. However, when amplification exceeds this range ($\alpha \geq 100$), the model fabricates information and produces repetitive or nonsensical output.

The Llama model (Figure 10) benefits from a narrower amplification range ($\alpha$ = 15-25). Within this window, responses become slightly more polite and helpful but lack substantive content. Beyond this range, output quickly becomes repetitive and loses coherence. Overall, Qwen improves output quality over a broader amplification range, while Llama's effective range is more limited.

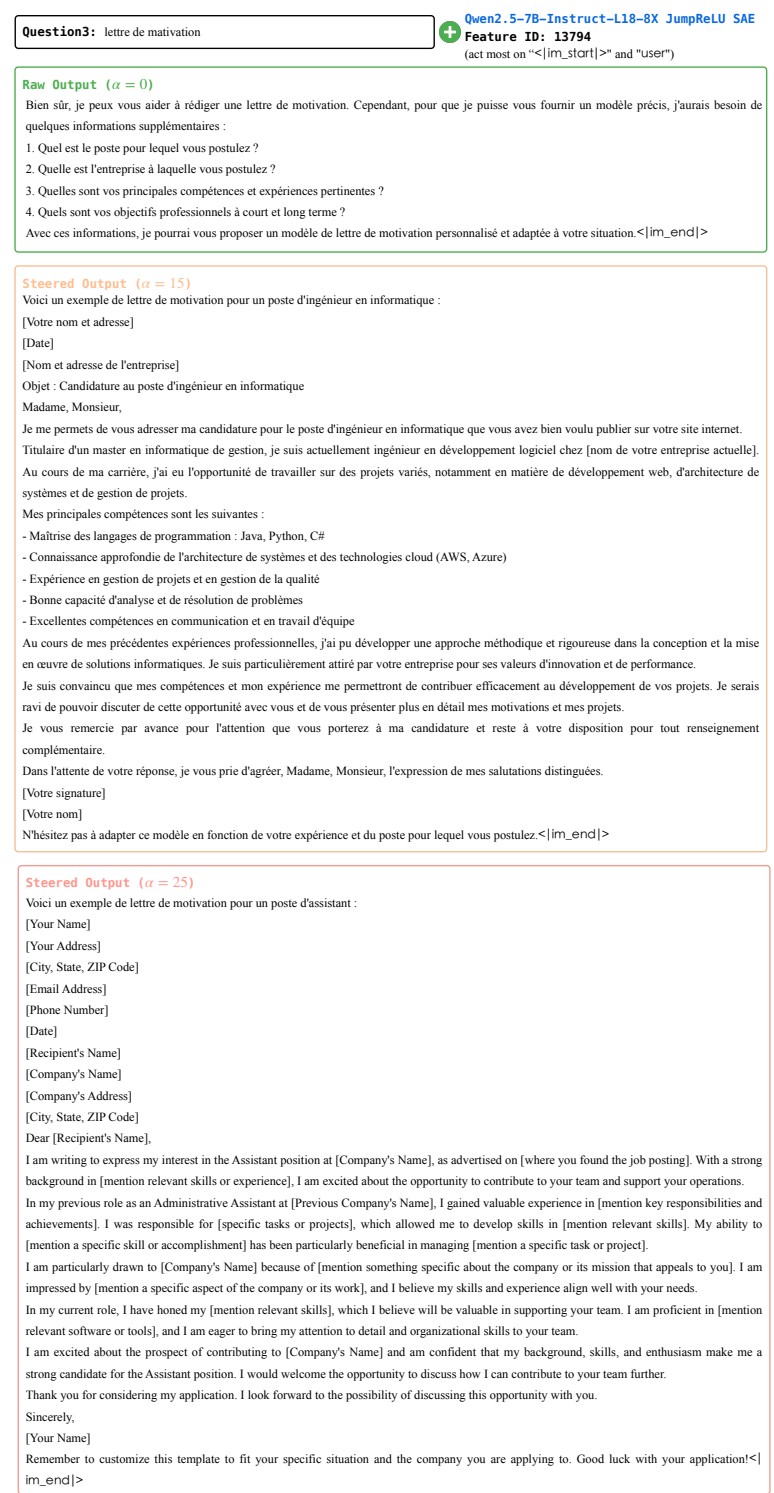

*Figure 11.* The steering output generated by Qwen2.5-7B-Instruct with Feature ID: 13794, focusing on `user` and `<|im_start|>` tokens for the Question 3 (cover letter task).

### M.4.2. Q3

For Question 3, the Qwen model (Figure 11) shows that moderate amplification of feature 13794 (with $\alpha$ between 50 and 100) leads to more informative and structured responses, providing richer content and clearer reasoning. This indicates a

substantial improvement in output quality within this coefficient range. However, further increasing the amplification causes the model to hallucinate, such as switching languages or generating irrelevant content, and ultimately results in repetitive or meaningless output.

The Llama model (Figure 12) also exhibits some improvement in informativeness and engagement when its most active feature is lightly amplified, but this effect is only present at very low coefficients (up to about $\alpha = 25$). Beyond this point, the output rapidly deteriorates into repetitive or off-topic text. Compared to Qwen, Llama's window for beneficial amplification is much narrower and less robust.

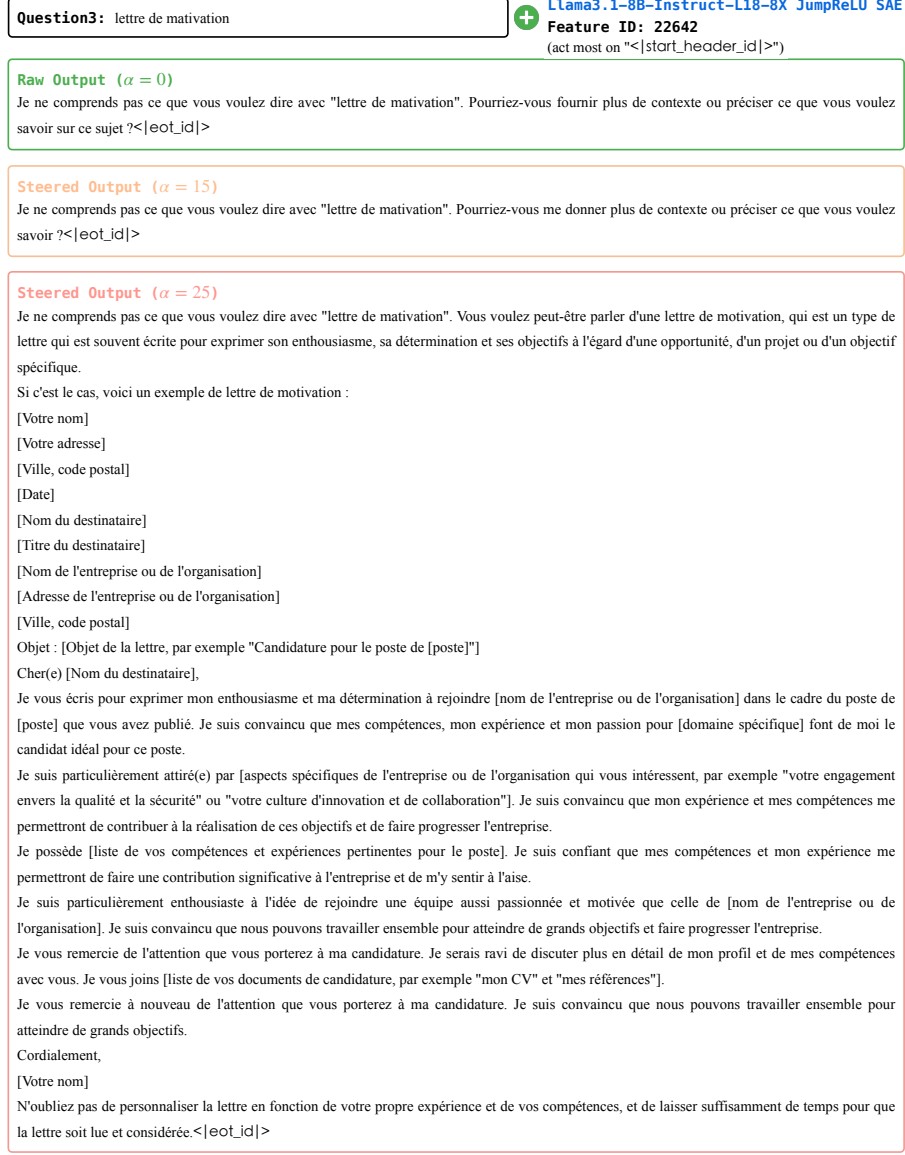

*Figure 12.* The steering output generated by Llama3.1-8B-Instruct with Feature ID: 22642, focusing on `<|start_header_id|>` tokens for the Question 3 (cover letter task).

M.4.3. Q4

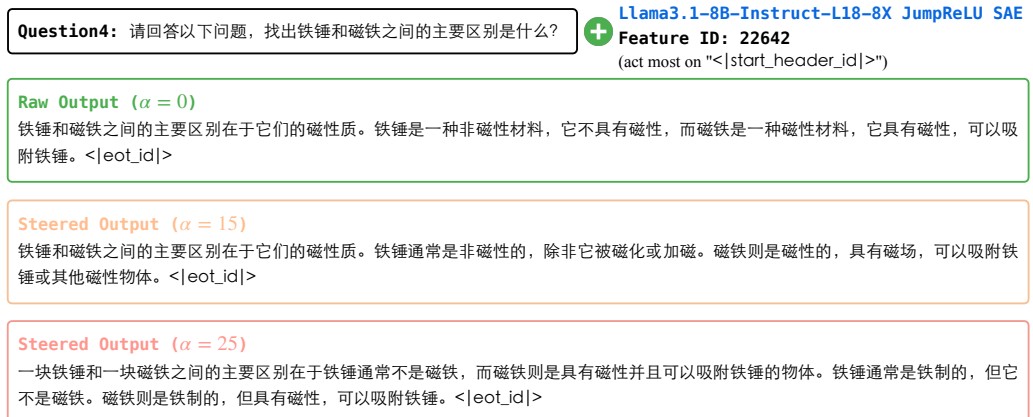

*Figure 13.* The steering output generated by Qwen2.5-7B-Instruct with Feature ID: 13794, focusing on `user` and `<|im_start|>` tokens for the Question 4 (entity discrimination task).

**Question4:** 请回答以下问题，找出铁锤和磁铁之间的主要区别是什么？

`Llama3.1-8B-Instruct-L18-8X JumpReLU SAE`
**Feature ID: 22642**
(act most on "`<|start_header_id|>`")

`Raw Output (α = 0)`
铁锤和磁铁之间的主要区别在于它们的磁性质。铁锤是一种非磁性材料，它不具有磁性，而磁铁是一种磁性材料，它具有磁性，可以吸附铁锤。<|eot_id|>

`Steered Output (α = 15)`
铁锤和磁铁之间的主要区别在于它们的磁性质。铁锤通常是非磁性的，除非它被磁化或加磁。磁铁则是磁性的，具有磁场，可以吸附铁锤或其他磁性物体。<|eot_id|>

`Steered Output (α = 25)`
一块铁锤和一块磁铁之间的主要区别在于铁锤通常不是磁铁，而磁铁则是具有磁性并且可以吸附铁锤的物体。铁锤通常是铁制的，但它不是磁铁。磁铁则是铁制的，但具有磁性，可以吸附铁锤。<|eot_id|>

*Figure 14.* The steering output generated by Llama3.1-8B-Instruct with Feature ID: 22642, focusing on `<|start_header_id|>` tokens for the Question 4 (entity discrimination task).

In Question 4, both models show that feature amplification can enhance Chain-of-Thought (CoT) (Wei et al., 2023) reasoning and answer quality, but only within specific coefficient ranges. For Qwen (Figure 13), amplifying the most active feature with $\alpha$ between 25 and 100 produces more convincing, informative, and well-structured responses. This improvement is especially evident in the quality of reasoning and the clarity of the final answers. However, excessive amplification again leads to a loss of coherence and informativeness.

For Llama (Figure 14), a similar pattern is observed but within an even narrower range. Mild amplification (up to $\alpha = 25$) can slightly improve the quality of reasoning and engagement, but any further increase quickly causes the output to become repetitive and less meaningful. This highlights that while both models benefit from feature amplification, Qwen maintains improved output quality over a wider range of coefficients, whereas Llama's useful range is much more restricted.

