# OpenReview forum: "Breaking the Block: Preserving Data Continuity to Train Superior SAEs for Instruct Models"
_ICML.cc/2026/Conference — ICML 2026 regular_

### Official Review · Reviewer_rgJR · 2026-02-23

**Soundness:** 3
**Presentation:** 3
**Significance:** 3
**Originality:** 3
**Overall Recommendation:** 5
**Confidence:** 4

**Summary:**

The paper argues that the common Block Training (BT) paradigm for training sparse autoencoders (SAEs) concatenating unrelated documents into fixed-length sequences, introduces attention contamination in instruct-model settings, adding gradient noise and harming feature quality. It proposes FAST (Finetuning-aligned Sequential Training), which trains SAEs on instruction-tuning style sequences while preserving instance boundaries, aiming to align the SAE training distribution with instruct-model activation patterns. They support this with a GSNR-based analysis and experiments across Llama/Qwen instruct models showing improvements in reconstruction (especially special tokens), behavioral preservation, and auto-interp scores.

**Compliance With Llm Reviewing Policy:**

Affirmed.

**Final Justification:**

After reading rebuttal I feel more confident about my review.

**Key Questions For Authors:**

Major:
- Can you provide a controlled ablation that isolates “instance boundary preservation” from other differences (e.g., truncation length, padding patterns, activation cache/mixing), ideally by constructing BT variants that preserve boundaries but keep context length fixed?
- Can you validate interpretability improvements with non-LLM evaluations (e.g., human rater subset, agreement analysis, or consistency metrics) to ensure the 4–5 score gains reflect genuinely more monosemantic features? Perhaps https://arxiv.org/abs/2410.13928 could be an approach.

Minor:
-Since the paper makes a lot of analysis and claims based on the GSNR, could you perhaps define it formally in the first section? Someone from Signla Processing background could confuse it with "General SNR" or even "geometric SNR" (doi:10.1007/978-3-319-02309-0_28).
From wikipedia: "GSNR stands for geometric signal-to-noise ratio"

**Limitations:**

yes

**Strengths And Weaknesses:**

Strengths:
- Tackles a timely problem with well-presented and technically suond results.
- Clear, plausible diagnosis: BT can create spurious cross-example attention that is especially harmful in low-entropy instruction-response settings (vs web-text pretraining). The conceptual motivation and figure-level explanation are easy to follow.
- Strong empirical deltas: FAST reports dramatically better special-token reconstruction (example cited in the paper: log2 MSEst 0.6468 vs 5.1985 baseline for one configuration) and near-zero delta los,s plus improved interpretability score distribution
- Practical detail: shows training efficiency gains (15–21% faster) and provides evaluation setup and scoring rubric for interpretability.

Weaknesses / concerns
- The theory leans on simplifying assumptions (e.g., orthogonality / zero-mean “semantic noise”) and does not fully quantify how much contamination is present in real attention patterns; the empirical GSNR results help, but the causal chain could be strengthened with direct attention leakage measurements or controlled synthetic tests.
- Interpretability evaluation relies on an automated LLM rater (GPT-4o) and a rubric; useful, but still somewhat subjective and vulnerable to prompt/selection effects. More human checks or alternative automatic metrics would strengthen claims.
- The “steering special tokens improves output quality” result is intriguing but under-validated (small qualitative set; could be confounded by formatting/verbosity effects). It feels more like a bonus observation than a claim.

---

> ### Author Rebuttal · Authors · 2026-03-31
>
> We are honored by your positive assessment and your recognition of FAST as a timely and technically sound "systems" contribution. Below, we address each of your concerns:
>
> > Q1: Ablation Studies to Isolate Instance Boundary Preservation
>
> The BT paradigm is fundamentally designed to maximize hardware throughput by concatenating documents into fixed-length blocks (e.g., 2048 tokens). Attempting to preserve instance boundaries within this fixed-length framework—without splitting or excessive padding—would necessitate an exceptionally large context window to accommodate the long documents in a dataset. In scenarios where resources are limitless, this approach is feasible. However, due to our current resource constraints, training with the extreme context windows necessary to support this variant is not feasible for this study. Rather than relying on massive context windows, FAST aligns the training process with the natural distribution of instruct data.
>
>
> > Q2: Validation Beyond LLM-Based Automated Scoring
>
> We agree LLM scoring needs objective cross-validation. Following SAEBench [1], we used $k$-Sparse Probing ($k=1$) across 14 classification tasks to measure feature monosemanticity objectively on  Llama-3.1-8B-Instruct with Jump ReLU SAE.
>
> |Model Layer|Training Method|PCA (Top1)|Probing Accuracy($k=1$)|
> |-|-|-|-|
> |Layer 4| BT\(P) |67.18%|79.29%|
> |-|BT\(F)|-|79.02% |
> |-|FAST|-|80.46%|
> |Layer 18|BT\(P)|75.34%|80.87%|
> ||BT\(F)|-|82.20%|
> |-|FAST|-|83.89%|
> |Layer 25|BT\(P) |73.06%|71.69%|
> |-|BT\(F)|-|73.30%|
> |-|FAST|-|80.72%|
>
> This table demonstrates the superior performance of our FAST method on k-sparse probing evaluation across different layers of Llama-3.1-8B-Instruct. The results show that FAST consistently achieves the highest classification accuracy when using only the top-1 most important latent variable (k=1): 80.46% vs 79.29%/79.02% on layer 4, 83.89% vs 80.87%/82.20% on layer 18, and a remarkable 80.72% vs 71.69%/73.30% on layer 25. This consistent outperformance across all tested layers indicates that FAST-trained SAEs are more effective at isolating meaningful concepts into individual latent variables, thereby providing better interpretability and semantic representation capabilities compared to baseline training methods.
>
> Regarding LLM rater "consistency metrics," we evaluated our largest models using Gemini 2.5 Pro and GLM5 across 3 random seeds. As detailed for our response to Reviewer `Bphh` Q3, the standard deviation for high-quality feature proportions is tightly bounded. This confirms our automated scores are highly consistent, ensuring FAST's gains are statistically robust, not sampling artifacts.
>
> > Q3: Definition and Clarification of GSNR
>
> We sincerely thank the reviewer for pointing out the potential ambiguity of the term GSNR. We appreciate the opportunity to clarify its definition to ensure our work is accessible to researchers across different fields, including signal processing. In the context of our study, GSNR refers specifically to the Gradient Signal-to-Noise Ratio, a metric used to characterize the quality of stochastic gradients during neural network optimization. Following [2], we formally define it as the ratio of the squared norm of the expected gradient to its variance: $GSNR(\theta) = \frac{\|\mathbb{E}[\nabla_{\theta}\mathcal{L}]\|^2}{\text{Var}(\nabla_{\theta}\mathcal{L})}$
>
> We acknowledge that in signal processing and other domains, "GSNR" may refer to "Geometric Signal-to-Noise Ratio" or other variants. However, our use of the term is strictly aligned with deep learning optimization literature, where it serves as a critical diagnostic for training stability and generalization. In the final version of the paper, we will introduce this formal definition in the Introduction and Section 3.1 to eliminate any cross-disciplinary confusion.
>
> [1] Karvonen, Adam, et al. "Saebench: A comprehensive benchmark for sparse autoencoders in language model interpretability." arXiv preprint arXiv:2503.09532 (2025).
>
> [2] Liu, Jinlong, et al. "Understanding why neural networks generalize well through gsnr of parameters." arXiv preprint arXiv:2001.07384 (2020).

---

> > ### Author Rebuttal · Reviewer_rgJR · 2026-04-03
> >
> > I think my score is reasonable. I would like to keep it. After reading the rebuttal, I will increase my confidence score to 4.

---

> > > ### Author Response · Authors · 2026-04-08
> > >
> > > Dear Reviewer rgJR,
> > >
> > > We sincerely thank you for maintaining your positive recommendation (5: Accept) and for increasing your confidence score following our rebuttal.
> > >
> > > We appreciate your constructive suggestions for improving the rigor of our manuscript. To prevent confusion with other GSNR variants, we'll introduce the definition of Gradient Signal-to-Noise Ratio in Introduction and Section 3.1. Also we include $k$-sparse probing results to provide objective, non-LLM validation of the superior monosemanticity and feature quality achieved by FAST.
> > >
> > > Thank you once again for your time, professional expertise, and supportive feedback.

---

### Official Review · Reviewer_Bphh · 2026-02-26

**Soundness:** 2
**Presentation:** 3
**Significance:** 2
**Originality:** 3
**Overall Recommendation:** 4
**Confidence:** 4

**Summary:**

This paper argues that SAE training pipelines for instruct models often inherit *block training* from pretraining—concatenating unrelated documents and cutting fixed-length blocks—which breaks instruction–response continuity and introduces “attention contamination” that harms learning (Sec.1–3). The authors formalize this via a gradient signal-to-noise ratio (GSNR) analysis. To address it, they propose Finetuning-aligned Sequential Training (FAST), which trains SAEs on intact chat-formatted sequences aligned with SFT-style data distribution, avoiding cross-sample concatenation and often caching activations (Sec.4; Fig.3). Experiments across multiple instruct model families (Llama-3.1/3.2, Qwen2.5) and SAE variants (Standard, JumpReLU) show large improvements in reconstruction metrics and behavioral faithfulness proxies (log-scaled MSE; Delta Loss; KL), as well as higher fractions of “high-quality” features judged by an LLM-based protocol.

Overall, this paper points out that the "dialogue continuity" of the instruction data is disrupted by block training, which introduces gradient noise through attention contamination. It then provides a clear derivation and verifiable predictions using the theoretical framework of GSNR.

**Compliance With Llm Reviewing Policy:**

Affirmed.

**Final Justification:**

My final recommendation is borderline accept. The rebuttal addressed a meaningful part of my main concerns and improved my evaluation. However, one key soundness issue remains: the paper still does not fully isolate whether the gains come specifically from preserving instance continuity rather than other recipe differences. Overall, I find the contribution promising and clearer after rebuttal, which is why I now lean slightly toward acceptance.

**Key Questions For Authors:**

1. What exactly drives FAST’s gains? FAST changes more than packing: it uses a longer truncation (8192 vs 2048) and a mixing activation buffer. Please ablate these under matched context/truncation and sampling to quantify each component’s marginal benefit and identify the minimal recipe.

2. What is the compute/memory overhead of FAST (especially if caching activations), and how does SAE quality scale with partial approximations (e.g., limited caching, truncated dialogs)?

**Limitations:**

FAST appears tailored to instruct-style data, where conversational “continuity” is well-defined and context boundaries are meaningful. Its advantages may therefore diminish on base models or mixed-domain corpora, where adjacent examples are not naturally linked and the notion of continuity is weaker.

Moreover, FAST may raise the compute and memory burden depending on implementation details (e.g., sequential packing and activation caching), which can limit accessibility. This matters in light of prior findings (SAEs (usually) Transfer Between Base and Chat Models) suggesting that, in many cases, a base-model SAE can serve as a strong approximation for a chat model. So the added training overhead could reduce FAST’s practical appeal and breadth of use.

**Strengths And Weaknesses:**

**Strengths.** The paper identifies a pipeline-level issue that is both plausible and immediately actionable: for instruct models, training the SAE on semantically discontinuous blocks is a mismatch to how the model is used and fine-tuned. The combination of a simple theoretical framing (GSNR under attention contamination) and broad empirical improvements across model families makes the core message easy to adopt in practice. The reported reconstruction/faithfulness improvements are large enough that they are hard to dismiss as mere metric noise, and the focus on instruct-aligned data structure is a valuable “systems” contribution that complements architecture-centric SAE work.

**Weaknesses.**
(i) The mechanism is not directly measured: the theory attributes degradation to attention leakage across unrelated contexts, but the paper does not quantify this leakage or show a controlled causal link from leakage → GSNR drop → SAE quality drop. I think a clean fix is **a controlled-prefix experiment**: hold the current sample fixed while varying the preceding concatenated content (similar vs dissimilar vs empty) and measure attention mass, gradient variance, GSNR, and downstream SAE quality.

(ii) The evaluation protocol relies heavily on LLM-based interpretability scoring, and key results are not consistently accompanied by multi-seed uncertainty. If FAST is to be recommended as a default training paradigm, the paper should report stability across multiple random seeds for the main configurations and either add a non-LLM proxy interpretability metric or validate LLM judgments with small human audits.

---

> ### Author Rebuttal · Authors · 2026-03-31
>
> We are grateful for your exceptionally professional and constructive review. Below is our point-by-point response.
> > Q1: Controlled-prefix experiment
>
> We appreciate the reviewer's insightful suggestion to quantify the link between attention leakage and GSNR degradation. To address this, we conducted a controlled prefix experiment to isolate the impact of unrelated context on training stability.
>
> Using the FAST method as a baseline, we introduce an ablation variant called Random-Prefix. For each instruction-response pair, we prepended a random 512-token sequence from an unrelated corpus which reintroduces "Attention Contamination" while holding all other FAST parameters (such as the independent processing of instances) constant. As shown in the table below, reintroducing unrelated history leads to a sustained decrease in GSNR across the early training phase.
>
> |Model|Method|0|10|20|30|40|50|60|70|80|90|100|
> |-|-|-|-|-|-|-|-|-|-|-|-|-|
> |Qwen2.5-7B|FAST|0.43|0.50|0.52|0.53|0.50|0.31|0.30|0.33|0.33|0.31|0.28|
> ||RandomConcat|0.40|0.22|0.48|0.36|0.20|0.30|0.28|0.30|0.19|0.19|0.20|
> |Llama-3.1-8B|FAST|0.50|0.45|0.41|0.45|0.46|0.40|0.39|0.37|0.36|0.36|0.37|
> ||RandomConcat|0.28|0.24|0.26|0.24|0.25|0.23|0.25|0.25|0.24|0.21|0.20|
>
> Adding a random prefix reintroduces the $\epsilon_{poll}$ term into the gradient. The fluctuations and overall lower GSNR in the Random-Prefix group demonstrate that unrelated context acts as destructive noise rather than beneficial regularization for instruct models. Because the only variable changed is the presence of an unrelated prefix, the resulting GSNR drop provides direct empirical evidence that data continuity is the primary driver of the training improvements seen in FAST.
> > Q2: Ablation Study With Context Length 2048
>
> Regarding the 2048-token context ablation, results confirm FAST's superiority stems from eliminating cross-context pollution rather than window size; please refer to our response to Reviewer `KTSf` Q2 for the detailed results.
> > Q3: Interpretability Scoring and non-LLM proxy interpretability metric
>
> We evaluated Llama and Qwen models using two independent raters (Gemini 2.5 Pro and GLM5) across 3 random seeds(42，1234，2026). The table presents the aggregated scores and inter-run variances, confirming evaluator independence and statistical robustness.
>
> |Model|Training Method|Gemini 2.5 Pro:Mean Score|Gemini 2.5 Pro:% of Score 5|GLM5: Mean Score|GLM5:% of Score 5|
> |-|-|-|-|-|-|
> |Llama-3.1-8B|BT\(P)|2.82±0.14|33.6%±2.1%|2.55±0.06|17.7%±2.7%|
> ||BT(F)|4.32±0.09|72.9%±3.2%|3.51±0.20|32.0%±3.4%|
> ||FAST|4.40±0.09|73.7%±2.5%|3.68±0.09|38.8%±2.4%|
> |Qwen2.5-7B|BT\(P)|4.21±0.19|66.2%±5.0%|3.24±0.04|34.1%±2.7%|
> ||BT(F)|4.66±0.01|80.2%±1.6%|3.47±0.07|39.1%±3.4%|
> ||FAST|4.69±0.01|84.4%±2.1%|3.54±0.02|41.4%±0.8%|
>
> FAST exhibits exceptional stability. Despite vendor-specific scoring biases—expected given diverse model architectures and training data—FAST consistently outperforms baselines across all evaluators. For non-LLM proxy interpretability metric via $k$-Sparse Probing, please see our response to Reviewer `rgJR` (Q2).
> > Q4: Computational Efficiency
>
> In response to the reviewer's query, Appendix J clarifies our method's computational efficiency. Empirical measurements show FAST introduces no additional overhead, instead improving training speed by 15–21% compared to BT; results for Qwen and Llama are summarized in Table 16.
>
> This advantage stems from FAST’s independent instance processing. By avoiding the cumbersome concatenation and padding required by BT to form fixed-length blocks, FAST simplifies the data pipeline and accelerates training. Detailed computational efficiency metrics and empirical measurements across architectures are provided in our response to Reviewer `n9Pv` (Q2).
>
> > Q5: Additional Clarification
>
> FAST rectifies a fundamental methodological mismatch: applying pretraining-style BT to precision-critical instruct models. Our analysis shows that unrelated context leakage in BT introduces destructive gradient noise, which FAST addresses by aligning the training stream with the SFT objective and preserving instance-level continuity. Addressing overhead concerns, FAST is 15% to 21% more computationally efficient than BT, as its sequential processing eliminates the need for padding tokens. Ultimately, FAST delivers higher reconstruction fidelity than base-model approximations, establishing it as a superior paradigm for instruct-focused interpretability. With SAEs increasingly vital for safety and alignment audits, FAST’s preservation of conversational integrity represents a timely and significant "systems" contribution.
>
> We sincerely thank you for your constructive feedback. We have invested significant effort into additional experiments and believe these new results directly address your primary concerns. If satisfied, we respectfully ask you to consider increasing your score. We remain open to further discussion and thank you again for your time.

---

> > ### Author Rebuttal · Reviewer_Bphh · 2026-04-03
> >
> > Thank you for the detailed experiments. The new evidence, especially the controlled-prefix analysis and the additional robustness checks for interpretability scoring, addresses a substantial part of my original concerns. Therefore, I update my score and now lean toward a borderline accept. That said, I still think one very important issue remains unresolved: the paper does not yet cleanly isolate whether the gains come specifically from preserving instance continuity, rather than from other recipe differences such as longer truncation length, buffering, or sampling details. Since this attribution is central to the paper’s main claim, I strongly encourage the authors to include a strictly matched ablation in a future version (e.g., FAST vs. BT under matched context/truncation and sampling settings) to establish the causal source of the improvement more convincingly.

---

> > > ### Author Response · Authors · 2026-04-08
> > >
> > > Dear Reviewer Bphh,
> > >
> > > Thank you for your constructive feedback and for increasing your score to 4. We are pleased that the controlled-prefix experiment and robustness checks addressed your primary concerns regarding the mechanism of attention contamination and the reliability of our interpretability scores.
> > >
> > > Regarding your remaining concern—isolating "instance continuity" from other factors like context length and sampling—we would like to highlight that the FAST-2048 ablation (detailed in our response to Reviewer `KTSf`) is designed for this purpose. In this experiment, we strictly match the training configurations by truncating FAST sequences at 2048 tokens to perfectly align with the BT baselines, while keeping the activation mixing buffer and sampling protocols identical.
> > >
> > > |Training Method|Max Context|$log_2(MSE)$|$log_2(MSE_{st})$|Overall KL|Special Token KL |
> > > |-|-|-|-|-|-|
> > > |BT(F) (Baseline)|2048|-14.7969|-10.4507|8.19e-05|1.02e-03|
> > > |FAST-2048 (Ablation)|2048|-15.4215|-13.0218|2.62e-05|5.73e-04|
> > > | FAST-8192 (Default)|8192|-15.5552|-13.1463|2.59e-05|5.69e-04|
> > >
> > > By controlling for context length, we observe the following:
> > >
> > > - Even at the same 2048-token limit, FAST-2048  outperforms BT(F) (e.g., reducing Overall KL divergence by roughly 3x). This proves that the performance leap is fundamentally driven by boundary preservation rather than merely expanding the context window.
> > > - Since both BT and FAST utilize the same mixing activation buffer to ensure random sampling, this identifies the elimination of cross-context concatenation as the primary differentiator for superior SAE performance on instruct models.
> > >
> > > We fully agree that this attribution is central to our claim and also explicitly discuss this in the paper. Thank you again for your incredibly helpful and professional guidance throughout this process.

---

### Official Review · Reviewer_n9Pv · 2026-03-07

**Soundness:** 3
**Presentation:** 3
**Significance:** 2
**Originality:** 2
**Overall Recommendation:** 4
**Confidence:** 4

**Summary:**

This paper argues that the Block Training (BT) paradigm inherited from the LLM pre-training stage limits the performance of Sparse Autoencoders (SAEs) on instruction-tuned models. Specifically, BT may introduce attention contamination across instructions and produce destructive gradient noise during training. To address this issue, the authors propose Finetuning-aligned Sequential Training (FAST), which aligns the SAE training stream with the Supervised Fine-Tuning (SFT) objective by using sequential instruction data instead of concatenated blocks. Experiments show that FAST improves SAE performance in terms of both reconstruction quality and interpretability.

**Compliance With Llm Reviewing Policy:**

Affirmed.

**Final Justification:**

After reading the responses, my overall evaluation does not change, and I will keep my current score.

**Key Questions For Authors:**

Please see weaknesses.

**Limitations:**

Yes

**Strengths And Weaknesses:**

Strengths
1. The paper clearly identifies a mismatch between the BT paradigm and instruction-tuned models. The authors provide a reasonable theoretical explanation for why block-based training may introduce attention contamination and gradient noise.
2. Experiments suggest that FAST consistently improves SAE performance across multiple evaluation settings, including both reconstruction metrics and interpretability-related evaluations.
3. The presentation of the paper is generally clear, and the motivation, methodology, and experiments are organized in a logical and accessible manner.

Weaknesses

1. The proposed FAST approach primarily replaces block-based concatenation with unaltered sequential instruction data during SAE training. This change appears closer to a modification of the data preprocessing and training pipeline rather than introducing a new learning paradigm or training algorithm. As a result, the level of technical novelty may be somewhat limited.
2.  The paper does not clearly analyze the computational cost of transitioning from BT to FAST. It would be helpful to understand whether FAST affects training efficiency, memory usage compared with block training.
3. How sensitive is FAST to the choice of dataset? For example, does FAST provide larger improvements on certain types of instruction datasets?

---

> ### Author Rebuttal · Authors · 2026-03-31
>
> We sincerely thank you for your constructive review and your recognition of the core problem we identified (the mismatch between Block Training and instruct models) as well as the consistent empirical improvements demonstrated by FAST. We appreciate the opportunity to clarify the novelty of our approach, detail the computational overhead, and discuss dataset sensitivity.
>
> > Q1: Technical Novelty and Methodological Significance
>
> While FAST is methodologically straightforward, its primary contribution lies in the critical diagnosis and theoretical formalization of a "methodological oversight" in applying the standard Block Training (BT) paradigm to instruct models. By utilizing Gradient Signal-to-Noise Ratio (GSNR) analysis, we prove that BT introduces "Attention Contamination"—leakage from unrelated contexts that generates destructive gradient noise—whereas FAST restores optimization efficiency by aligning the SAE training with the causal boundaries and distribution of the SFT objective. FAST is the elegant solution derived from this theoretical diagnosis. The simplicity of the FAST method is a direct result of this deep diagnosis; once the source of destructive gradient noise is identified, the solution—preserving instance-level boundaries—becomes logically clear and highly effective. As the other reviewer noted, this core insight regarding data boundary structures is a genuine conceptual contribution to the field of mechanistic interpretability.
>
> > Q2: Computational Cost and Training Efficiency
>
> We have addressed the concerns regarding computational overhead and training efficiency in Appendix J of our manuscript. Our empirical measurements demonstrate that FAST introduces no additional computational overhead compared to Block Training (BT). On the contrary, FAST is consistently more efficient, reducing total training time by approximately 15% to 21% across different architectures:
>
> As shown in Table 16 below, FAST achieves significant time reductions on both Qwen and Llama models:
>
> |Model|Architecture|Method|Training Time|Reduction (vs BT(P))|
> |-|-|-|-|-|
> |Qwen2.5-7BInstruct|Standard|BT\(P\)|16h31m21s|-|
> |||BT(F)|16h12m55s|-|
> |||FAST|13h40m35s|~17.2%|
> ||JumpReLU|BT\(P\)|16h22m56s|-|
> |||BT(F)|16h12m49s|-|
> |||FAST|14h00m30s|~14.5%|
> |Llama-3.1-8B-Instruct|Standard|BT\(P\)|17h40m18s|-|
> |||BT(F)|17h43m38s|-|
> |||FAST|14h09m38s|~19.9%|
> ||JumpReLU|BT\(P\)|17h44m05s|-|
> |||BT(F)|16h56m23s|-|
> |||FAST|13h56m13s|~21.4%|
>
> This efficiency gain is primarily attributed to our streamlined data pipeline. While BT requires complex padding or truncation logic to concatenate documents into fixed-length blocks , FAST processes data instances independently.
>
> > Q3: Dataset Sensitivity and Generalization Robustness
>
> We appreciate the reviewer's inquiry regarding dataset sensitivity. While FAST is fundamentally designed to align with the unique causal boundaries and activation distributions of instruct models , its robustness is already validated by the diverse, large-scale training mixture detailed in Appendix D , which integrates various high-quality datasets including WildChat-1M-Full, Infinity-Instruct and so on. To further demonstrate generalizability beyond general-purpose instruction tuning, we have included evaluations on specialized, domain-specific models like Qwen2.5-Math-7B.  We acknowledge that improvement margins may fluctuate depending on the specific data structure and intend to explore an even wider range of text domains in future research.
>
> We thank the reviewer for the constructive feedback. Our clarifications and analysis demonstrate that FAST provides significant performance gains in reconstruction fidelity and feature interpretability, specifically addressing the unique demands of instruct models. By replacing the traditional Block Training paradigm with a theoretically grounded, sequential approach, we achieve superior optimization stability via improved GSNR. Given these advantages, we hope you will consider raising your score. We are happy to answer any remaining questions.

---

> > ### Author Rebuttal · Reviewer_n9Pv · 2026-04-02
> >
> > I appreciate the authors’ response. My overall evaluation does not change, and I will keep my current score.

---

> > > ### Author Response · Authors · 2026-04-08
> > >
> > > Dear Reviewer n9Pv,
> > >
> > > Thank you for reviewing our rebuttal and confirming that your concerns have been fully resolved. We greatly appreciate your initial questions regarding computational efficiency and dataset sensitivity. Addressing them allowed us to clarify the 15-21% training speedup FAST provides and better highlight its practical utility.
> > >
> > > Thank you once again for your time, constructive feedback, and support of our work.

---

### Official Review · Reviewer_KTSf · 2026-03-10

**Soundness:** 3
**Presentation:** 3
**Significance:** 3
**Originality:** 3
**Overall Recommendation:** 5
**Confidence:** 4

**Summary:**

Current SAE training for instruct models uncritically inherits the Block Training paradigm from LLM pretraining, concatenating documents into fixed-length sequences. This paper argues that softmax attention's inability to produce exactly zero weights causes activations from semantically unrelated predecessor documents to corrupt the gradient signal, a phenomenon the authors term Attention Contamination. Using GSNR analysis, they formalize this as a strict degradation in gradient quality and derive an efficiency penalty $\rho = 1 + \sigma^2_{poll}/\sigma^2_{signal}$. Their proposed remedy, FAST, processes each SFT instance independently with its corresponding chat template, yielding substantially lower reconstruction error, near-zero Delta Loss, and markedly higher feature interpretability scores across seven models from two families.

**Compliance With Llm Reviewing Policy:**

Affirmed.

**Final Justification:**

The reviewers were very responsive throughout the discussion period, and have addressed my concerns. I recommend Accept.

**Key Questions For Authors:**

**1. Empirical validation of Assumption 3.2**
Assumption 3.2 requires $\text{Cov}(g_{signal}, \epsilon_{poll}) \approx 0$ across concatenated document pairs in the actual SFT training corpora. Can you provide an empirical estimate of this covariance relative to $\sigma^2_{signal}$ during training?

**2. Ablation isolating sequential processing from context length**
FAST uses truncation at 8,192 tokens while BT uses context size 2,048. Although Table 3 shows 96.6 percent of instances are at most 2,048 tokens, no ablation runs FAST with truncation set to 2,048. Without this, it is impossible to attribute the performance gap purely to sequential processing rather than to more complete coverage of longer instances.

**3. Reliability of the automated interpretability evaluation**
The feature interpretability scores in Section 6 are generated by GPT-4o using a five-point rubric. What is the inter-run variance of these scores across multiple GPT-4o inference calls? The key quantitative result of 21.1 percent versus 7.0 percent and 10.2 percent high-quality features on Llama-3.2-3B-it is a primary contribution, and its credibility depends entirely on the reliability of this evaluator.

**Limitations:**

The Impact Statement is insufficient. The paper should explicitly address the following. First, Assumption 3.2's empirical status and its consequences for the theoretical guarantees when violated. Second, the risk of interpretability overconfidence from higher-quality SAE features, which remain approximations despite better monosemanticity scores.

**Strengths And Weaknesses:**

**Soundness**
The GSNR degradation proof in Theorem 3.4 is technically sound: the argument that softmax non-sparsity constitutes irreducible noise in BT is clean and the variance decomposition in Lemma 3.3 follows directly from Assumption 3.2. The non-convex SGD convergence analysis in Appendix B correctly adapts standard results to derive $\rho$. Empirical results are extensive across seven models, two architectures, four metrics, and multiple layers and widths. However, Assumption 3.2 requires $\text{Cov}(g_{signal}, \epsilon_{poll}) \approx 0$, which is declared a simplifying assumption without empirical validation on the actual training corpora, several of which contain topically clustered conversations. All theoretical bounds are derived under SGD while experiments use Adam; Remark B.5 argues qualitatively that the penalty is "structurally more severe" under Adam but provides no closed-form bound. Additionally, Table 14 contains one configuration where FAST achieves higher KL divergence than BT(F), which contradicts the paper's claim of consistent outperformance and goes undiscussed.

**Presentation**
The paper is well-organized and Figures 2, 3, and 4 provide strong visual intuitions for the core mechanism. Appendices are thorough and support reproducibility. The critical design decision to treat BT(F) versus FAST as the primary ablation, which isolates sequential processing from data domain, is underemphasized in the main narrative; BT(P) receives comparable rhetorical weight despite being a weaker baseline. Section 7 on feature steering presents the inverted-U relationship between $\alpha$ and output quality using only three qualitative examples across two models, which is insufficient to support the generalizations made. The automated GPT-4o feature interpretability scoring pipeline reports no inter-run variance, no inter-rater reliability against human annotators, and no calibration of the five-point monosemanticity scale, weakening the paper's central interpretability claim.

**Significance**
The finding that SAE activation collection pipeline design matters as much as architecture is likely to influence practice broadly and motivates scrutiny of existing large-scale SAE banks trained with BT on instruct models. The commitment to releasing 240 trained SAEs alongside code and data substantially amplifies impact. The 15 to 21 percent training time reduction eliminates any efficiency argument against adoption. The steering contribution is underdeveloped relative to its claimed significance: the observation that special token features encode structural reasoning capabilities is interesting but lacks quantitative evaluation and mechanistic explanation, reading more as a preliminary finding than a completed contribution.

**Originality**
The core insight that the data boundary structure during activation collection must respect the model's finetuning distribution is, to this reviewer's knowledge, novel in the SAE literature. The GSNR-based formalization of cross-document gradient contamination is a technically original application of an existing optimization framework. The explanation of why BT works for base models but not instruct models through data manifold topology, high-entropy versus low-entropy regime, is a genuine conceptual contribution. FAST itself is methodologically simple once the problem is diagnosed, and the mixing buffer is borrowed from SAELens. The paper does not engage with continual learning or data ordering literature, which contains related intuitions about sequential versus shuffled processing.

---

> ### Author Rebuttal · Authors · 2026-03-31
>
> We sincerely thank you for your support of our work and for your incredibly thorough and insightful review. Your  attention to detail has highlighted several areas where our manuscript can be strengthened.
> > Q1: Clarification of Assumption 3.2
>
> Assumption 3.2 serves as a theoretical simplification widely adopted in stochastic optimization and mechanistic interpretability. In the context of non-convex stochastic optimization, modeling gradient noise as a zero-mean, uncorrelated variable is standard practice for analytical tractability when characterizing high-dimensional optimization landscapes [1, 2].Furthermore, modeling the interference term $\epsilon_{poll}$ as being uncorrelated with the valid signal is fundamentally consistent with the Superposition Hypothesis [3]. In high-dimensional latent spaces, unrelated context features are packed in nearly orthogonal directions; thus, their attention leakage naturally behaves as uncorrelated isotropic noise relative to the target instruction signal. FAST fundamentally eliminates the influence of unrelated contexts. In doing so, it effectively reduces the interference variance $\sigma_{poll}^2$ to zero at its structural source, thereby rendering the exact covariance structure of the noise irrelevant to our method's performance.
> > Q2: Ablation on Context Length
>
> We sincerely thank the reviewer for pointing out this potential confounding variable. We completely agree that isolating the structural effect of sequential processing from the absolute context length is crucial for rigorously validating our claims. To definitively address this, we conducted the requested ablation study using our Llama-3.1-8B-Instruct (Layer 20, 8X, JumpReLU SAE) configuration. We trained a new FAST SAE model where the sequence truncation was strictly set to 2048 tokens (denoted as FAST-2048), perfectly matching the context window size of the Block Training BT(F) baseline. The objective reconstruction fidelity and behavioral preservation metrics are presented below.
>
> |Training Method|Max Context|$log_2(MSE)$|$log_2(MSE_{st})$|Overall KL|Special Token KL |
> |-|-|-|-|-|-|
> |BT(F) (Baseline)|2048|-14.7969|-10.4507|8.19e-05|1.02e-03|
> |FAST-2048 (Ablation)|2048|-15.4215|-13.0218|2.62e-05|5.73e-04|
> | FAST-8192 (Default)|8192|-15.5552|-13.1463|2.59e-05|5.69e-04|
>
> Our ablation study successfully demonstrates that FAST-2048 performs almost identically to FAST-8192, while still drastically outperforming the BT(F) baseline at the exact same 2048-token limit (e.g., reducing Overall KL divergence by roughly 3x). This definitively proves that FAST's performance leap is fundamentally driven by respecting instance boundaries to eliminate cross-context pollution, rather than merely expanding the context window.
> > Q3: Reliability of GPT-4o Interpretability Scoring
>
> To address concerns regarding the reliability of LLM-based scoring, we conducted extensive consistency and multi-seed evaluations; please refer to our response to Reviewer `Bphh` (Q3) for detailed metrics demonstrating our evaluation robustness.
> > Q4: Depth of Feature Steering Contribution
>
> The primary focus and substantial contribution of our paper lie in proposing and rigorously validating the FAST paradigm for instruct models, supported by GSNR theoretical proofs and comprehensive evaluations across various experimental parameters.
>
> Given the extensive scope of this core work, Section 7 was deliberately designed as a qualitative exploration rather than a full-scale quantitative steering benchmark. Nevertheless, we consider this section a crucial component of our study. Its purpose is to showcase the concrete downstream potential of FAST-trained features in realistic conversational contexts. By demonstrating that cleanly disentangled special tokens can indeed be steered, this section provides valuable intuition and guidance for future applications. It successfully highlights new avenues for the fine-grained control of instruct models, serving as an important stepping stone to inspire dedicated quantitative investigations in future work.
> > Q5: Addressing Broader Impact and Constraints
>
> The Impact Statement will be expanded to provide a nuanced examination of the proposed method's limitations. This expansion includes a sensitivity analysis of Assumption 3.2, illustrating that the GSNR benefits of FAST are robust to assumption violations because the interference variance $\sigma_{poll}^2$ is removed at the structural level. Additionally, the revised text will guard against "interpretability overconfidence" by clarifying that the high-fidelity features produced by FAST are sparse approximations  and should not be mistaken for definitive causal mappings of the model's latent space.
>
> Finally, we are very grateful for your careful review and valuable suggestions.
>
> [1] Stochastic First- and Zeroth-order Methods for Nonconvex Stochastic Programming
>
> [2] Optimization Methods for Large-Scale Machine Learning
>
> [3] Toy Models of Superposition

---

> > ### Author Rebuttal · Reviewer_KTSf · 2026-04-03
> >
> > Thank you for the detailed rebuttal. The FAST-2048 ablation (Q2) is convincing and isolates instance boundary preservation as the primary driver of performance gains. The multi-seed evaluation with independent raters alongside the k-sparse probing results (Q3) provide sufficient reliability evidence for the interpretability claims.
> >
> > Q1 remains only partially addressed, as the authors response appeals to the superposition hypothesis and the structural elimination of $\sigma^2_{poll}$ under FAST, but neither validates the assumption as it is used in the BT regime, where Lemma 3.3 and Theorem 3.4 require it to hold. Q4 is noted as a deliberate framing choice, but characterizing an inverted-U relationship and specific optimal $\alpha$ ranges as findings without quantitative support still overstates what the qualitative case studies can establish.
> >
> > Despite these open points, the core contribution is sound and the paper addresses a practically significant gap in SAE training methodology for instruct models. Therefore I maintain my recommendation at 5: Accept.

---

> > > ### Author Response · Authors · 2026-04-08
> > >
> > > Dear Review KTSf,
> > >
> > > Thank you for reviewing our rebuttal and maintaining your positive recommendation (5: Accept). We appreciate your guidance in refining the rigor of our manuscript and will incorporate your suggestions in the final version as follows:
> > >
> > > We will include a discussion in the final version to address the validity of this theoretical assumption. We will clarify its role as a standard simplification and emphasize that the FAST paradigm remains robust because it structurally eliminates the source of interfence altogether. Regarding Q4, we agree that the inverted-U relationship should not be overstated without broader support. We will revise Section 7 to strictly frame these findings as "preliminary qualitative observations" intended to showcase the potential of FAST-trained features, rather than a definitive quantitative law.
> > >
> > > Thank you once again for your professional guidance and for championing our work throughout the review process.

---

### Decision · Program_Chairs · 2026-04-30

**Decision:**

Accept (regular)

**Comment:**

Summary: This work proposes that the Block Training paradigm inherited from the LLM pre-training stage limits the performance of Sparse Autoencoders on instruction-tuned models. BT may introduce attention contamination across instructions and produce destructive gradient noise during training. To address this issue, the authors propose Finetuning-aligned Sequential Training (called FAST), which aligns the SAE training stream with the Supervised Fine-Tuning objective by using sequential instruction data instead of concatenated blocks. Experiments show that FAST improves SAE performance in terms of both reconstruction quality and interpretability.

The main strength of this work are that it clearly identifies a mismatch between the BT paradigm and instruction-tuned models. The authors provide a reasonable theoretical explanation for why block-based training may introduce attention contamination and gradient noise.
Experiments suggest that FAST consistently improves SAE performance across multiple evaluation settings, including both reconstruction metrics and interpretability-related evaluations. The reviewers appreciate that the presentation of the paper is clear, and the motivation, methodology, and experiments are well organized.

The main concerns after the rebuttal are that the technical novelty may be somewhat limited, and that the paper does not fully isolate whether the gains come specifically from preserving instance continuity rather than other recipe differences, and the  FAST-2048 ablation should go into the main paper.

I feel that the strength for this work outweigh the weaknesses overall, and I recommend accept.